# Reward-Free Model-Based Reinforcement Learning with Linear Function Approximation

**Weitong Zhang**
Department of Computer Science
University of California, Los Angeles
Los Angeles, CA 90095
weightzero@cs.ucla.edu

**Dongruo Zhou**
Department of Computer Science
University of California, Los Angeles
Los Angeles, CA 90095
drzhou@cs.ucla.edu

**Quanquan Gu**
Department of Computer Science
University of California, Los Angeles
Los Angeles, CA 90095
qgu@cs.ucla.edu

## Abstract

We study the model-based reward-free reinforcement learning with linear function approximation for episodic Markov decision processes (MDPs). In this setting, the agent works in two phases. In the exploration phase, the agent interacts with the environment and collects samples without the reward. In the planning phase, the agent is given a specific reward function and uses samples collected from the exploration phase to learn a good policy. We propose a new provably efficient algorithm, called UCRL-RFE under the Linear Mixture MDP assumption, where the transition probability kernel of the MDP can be parameterized by a linear function over certain feature mappings defined on the triplet of state, action, and next state. We show that to obtain an $\epsilon$-optimal policy for arbitrary reward function, UCRL-RFE needs to sample at most $\widetilde{\mathcal{O}}(H^5 d^2 \epsilon^{-2})$ episodes during the exploration phase. Here, $H$ is the length of the episode, $d$ is the dimension of the feature mapping. We also propose a variant of UCRL-RFE using Bernstein-type bonus and show that it needs to sample at most $\widetilde{\mathcal{O}}(H^4 d(H + d)\epsilon^{-2})$ to achieve an $\epsilon$-optimal policy. By constructing a special class of linear Mixture MDPs, we also prove that for any reward-free algorithm, it needs to sample at least $\widetilde{\Omega}(H^2 d\epsilon^{-2})$ episodes to obtain an $\epsilon$-optimal policy. Our upper bound matches the lower bound in terms of the dependence on $\epsilon$ and the dependence on $d$ if $H \geq d$.

## 1 Introduction

In reinforcement learning (RL), the agent sequentially interacts with the environment and receives reward from it. In many real-world RL problems, the reward function is manually designed to encourage the desired behavior of the agent. Thus, engineers have to change the reward function time by time and train the agent to check whether it has achieved the desired behavior. In this case, RL algorithms need to be repeatedly executed with different reward functions and are therefore sample inefficient or even intractable. To tackle this challenge, Jin et al. [9] proposed a new reinforcement learning paradigm called *Reward-Free Exploration* (RFE), which explores the environment without using any reward function. In detail, the reward-free RL algorithm consists of two phases. The first phase is called *Exploration Phase*, where the algorithm explores the environment without receiving reward signals. The second phase is called *Planning Phase*, where the algorithm is given a specific

35th Conference on Neural Information Processing Systems (NeurIPS 2021).

reward function and use the collected data in the first phase to learn the policy. They have shown that this exploration paradigm can learn a near-optimal policy in the planning phase given *any* reward function after collecting polynomial number of episodes in the exploration phase. Follow up work [12, 14, 28] proposed improved algorithms to achieve better or nearly optimal sample complexity.

All the aforementioned works are focused on the tabular Markov decision process (MDP), where the number of states and actions are finite. In practice, the number of states and actions can be large or even infinite, and therefore *function approximation* is required for the sake of computational tractability and generalization. However, the understanding of function approximation for reward-free exploration, even under the simplest linear function approximation, remains underexplored, with only two notable related works [18, 27]. Specifically, Wang et al. [18] studied *linear MDPs* [21, 10], where both the transition probability and the reward function admit linear representations, and proposed a reward-free RL algorithm with $\widetilde{\mathcal{O}}(d^3 H^6 \epsilon^{-2})$ sample complexity, where $d$ is the dimension of the linear representation, $H$ is the planning horizon, and $\epsilon$ is the required accuracy. They also proved that if the optimal state-action function is linear, then the reward-free exploration needs an exponential number of episodes in the planning horizon $H$ to learn a $\epsilon$-optimal policy. Zanette et al. [27] considered a slightly larger class of MDPs with *low inherent Bellman error* [26], and proposed an algorithm with $\widetilde{\mathcal{O}}(d^3 H^5 \epsilon^{-2})$ sample complexity. However, both works assume the reward function is a linear function over some feature mapping. Moreover, the lower bound proved in [18] is for a very large class of MDPs where the optimal state-action function is linear, thus it is too conservative and cannot tell the information-theoretic limits of reward-free exploration for linear MDPs or related models.

In this paper, we seek a better understanding of the statistical efficiency for reward-free RL with linear function approximation. We propose two reward-free model-based RL algorithms for the finite-horizon episodic *linear mixture/kernel MDP* [16, 7, 3, 30], where the transition probability kernel is a linear mixture model. In detail, our contributions are highlighted as follows:

- We propose a new exploration-driven reward function and its corresponding pseudo value function for linear mixture MDPs, which will encourage the algorithm to explore the state-action pair with more uncertainty on the transition probability.

- We propose a UCRL-RFE algorithm which guides the agent to explore the state space using the exploration-driven reward function and pseudo value functions. We prove an $\widetilde{\mathcal{O}}(H^5 d^2 \epsilon^{-2})$ sample complexity for UCRL-RFE to achieve an $\epsilon$-optimal policy for any reward function for time-homogeneous MDP.

- We further propose a UCRL-RFE+ algorithm which uses a Bernstein-type exploration bonus. UCRL-RFE+ can reduce the error caused by the exploration-driven reward function during the exploration phase. With a novel analysis based on total variance, we prove an $\widetilde{\mathcal{O}}(H^4 d(H + d)\epsilon^{-2})$ sample complexity for UCRL-RFE+, which improves that of UCRL-RFE by a factor of $\min\{H, d\}$.

- By constructing a special class of linear mixture MDPs, we show that any reward-free algorithm needs to sample at least $\widetilde{\Omega}(H^2 d\epsilon^{-2})$ episodes to achieve an $\epsilon$-optimal policy for any reward function. This lower bound matches the upper bound of UCRL-RFE+ in terms of the dependence on the accuracy $\epsilon$ and feature dimension $d$ when $H \geq d$.

**Notation.** Scalars and constants are denoted by lower and upper case letters, respectively. Vectors are denoted by lower case bold face letters $\mathbf{x}$, and matrices by upper case bold face letters $\mathbf{A}$. We denote by $[k]$ the set $\{1, 2, \cdots, k\}$ for positive integers $k$. For two non-negative sequence $\{a_n\}, \{b_n\}$, $a_n = \mathcal{O}(b_n)$ means that there exists a positive constant $C$ such that $a_n \leq Cb_n$, and we use $\widetilde{\mathcal{O}}(\cdot)$ to hide the log factor in $\mathcal{O}(\cdot)$; $a_n = \Omega(b_n)$ means that there exists a positive constant $C$ such that $a_n \geq Cb_n$, and we use $\widetilde{\Omega}(\cdot)$ to hide the log factor. $a_n = o(b_n)$ means that $\lim_{n\to\infty} a_n/b_n = 0$. We denote by $S, A$ as the cardinality of the state set $\mathcal{S}$ and action set $\mathcal{A}$ separately. For a vector $\mathbf{x} \in \mathbb{R}^d$ and corresponding matrix $\mathbf{A} \in \mathbb{R}^{d \times d}$, we define $\|\mathbf{x}\|_{\mathbf{A}}^2 = \mathbf{x}^\top \mathbf{A} \mathbf{x}$. We denote $[x]_{(0,H)} := \max\{\min\{x, H\}, 0\}$. For vector $\mathbf{x} \in \mathbb{R}^d$, we denote by $[\mathbf{x}]_i$ the $i$-th element of $\mathbf{x}$.

## 2   Related Work

**Reinforcement Learning with Function Approximation.**   Function approximation is extremely useful for RL when the state space and/or the action space are large or even infinite. To develop

provable RL algorithms with linear function approximation, linear MDPs [10] is probably the most widely assumed MDP model, where both the transition kernel and the reward function are linear functions of a given feature mapping. A line of works has developed RL algorithms with polynomial sample complexity or regret bounds under this setting, such as LSVI-UCB [19] and randomized LSVI [24]. Besides the linear MDP, linear mixture/kernel MDPs [16, 7, 3, 30] has emerged as a new model which enables efficient RL with linear function approximation. In this setting, the transition kernel is a linear function over a feature mapping on the triplet of state, action, and next-state. Under this assumption, nearly minimax optimal regrets can be attained for both finite-horizon episodic MDPs and infinite-horizon discounted MDPs [29]. Note also that linear mixture MDPs do not require the reward function to be linear and therefore enables RL with arbitrary reward functions. Therefore, we also consider linear mixture MDPs in this paper.

| Algorithm | Sample Complexity | Time Homo. | MDP Type | Model Based |
|---|---|---|---|---|
| Jin et al. [9] | $\widetilde{\mathcal{O}}(H^5 S^2 A \epsilon^{-2})$ | $\times$ | Tabular | $\checkmark$ |
| RF-UCRL [12] | $\widetilde{\mathcal{O}}(H^4 S^2 A \epsilon^{-2})$ | $\times$ | Tabular | $\checkmark$ |
| RF-Express [14] | $\widetilde{\mathcal{O}}(H^3 S^2 A \epsilon^{-2})$ | $\times$ | Tabular | $\checkmark$ |
| SSTP [28] | $\widetilde{\mathcal{O}}(H^2 S^2 A \epsilon^{-2})$ | $\checkmark$ | Tabular | $\checkmark$ |
| Lower bound [9] | $\Omega(H^2 S^2 A \epsilon^{-2})$ | $\checkmark$ | Tabular | $\checkmark$ |
| Wang et al. [18] | $\widetilde{\mathcal{O}}(H^6 d^3 \epsilon^{-2})$ | $\times$ | Linear MDP | $\times$ |
| FRANCIS [26] | $\widetilde{\mathcal{O}}(H^5 d^3 \epsilon^{-2})$ | $\checkmark$ | Linear MDP | $\times$ |
| UCRL-RFE (Alg. 2) | $\widetilde{\mathcal{O}}(H^5 d^2 \epsilon^{-2})$ | $\checkmark$ | Linear Mixture | $\checkmark$ |
| UCRL-RFE+ (Alg. 3) | $\widetilde{\mathcal{O}}(H^4 d(H+d) \epsilon^{-2})$ | $\checkmark$ | Linear Mixture | $\checkmark$ |
| Lower bound (Thm. 6.1) | $\widetilde{\Omega}(H^2 d \epsilon^{-2})$ | $\checkmark$ | Linear MDP/Linear Mixture | $\checkmark$ |

Table 1: Comparison of episodic reward-free RL algorithms. *Time Homo.* stands for the MDP is a time-homogeneous, where the transition probabilities are the same at different stages of the episode. *Model Based* stands for the algorithm is a model-based algorithm ($\checkmark$) or a model-free algorithm ($\times$).

**Reward-Free Exploration.**   As the first work on reward-free exploration, Jin et al. [9] assigned each state an exploration-driven reward function at each round to guide the algorithm to do exploration. Then they utilized the EULER [25] algorithm to minimize the total regret. Their algorithm achieves an $\widetilde{\mathcal{O}}(S^2 A H^5 \epsilon^{-2})$ sample complexity in the tabular setting to achieve an $\epsilon$-optimal policy, where $S$ is the number of states and $A$ is the number of actions. They also proved a sample complexity lower bound as $\widetilde{\Omega}(S^2 A H^2 \epsilon^{-2})$. Kaufmann et al. [12] extended the UCRL [2] algorithm to the reward-free exploration. Their algorithm RF-UCRL achieves a sample complexity of $\widetilde{\mathcal{O}}(S^2 A H^4 \epsilon^{-2})$, which improves that of [9] by a factor of $H$. Ménard et al. [14] proposed RF-Express algorithm by modifying the UCB-bonus of UCRL to making it decay faster and achieved a sample complexity of $\widetilde{\mathcal{O}}(S^2 A H^3 \epsilon^{-2})$. Zhang et al. [28] proposed SSTP algorithm in the time-homogeneous setting, which achieves $\widetilde{\mathcal{O}}(S^2 A H^2 \epsilon^{-2})$ sample complexity, and matches the minimax lower bound provided in [9] up to logarithmic factors. Liu et al. [13] has shown the similarity between the self-play setting and reward-free setting. All of these works are for tabular MDPs.

Here we summarize and compare the related works on Reward Free Exploration in Table 1. Notice that our lower bound $\Omega(H^2 d \epsilon^{-2})$ for linear mixture MDPs can imply the same lower bound for linear MDPs and MDPs with low inherent Bellman error, using a similar argument used in [29].

## 3   Preliminaries

We consider episodic Markov Decision Processes (MDP), which is denoted by a tuple $M(\mathcal{S}, \mathcal{A}, H, \{r_h\}_{h=1}^{H}, \mathbb{P})$. Here $\mathcal{S}$ is the countable state space (may be infinite), $\mathcal{A}$ is the action space, $H$ is the length of the episode, $r_h : \mathcal{S} \times \mathcal{A} \to [0, 1]$ is the reward function. Without loss of generality, we assume the reward function $r_h$ is *deterministic*. $\mathbb{P}(s'|s, a)$ is the transition probability function which denotes the probability for state $s$ to transit to state $s'$ given action $a$ at step $h$. A policy $\pi_h : \mathcal{S} \to \mathcal{A}$ is a function which maps a state $s$ to an action $a$. We define the action-value

function (i.e., Q-function) $Q_h^\pi(s, a)$ as follows:

$$Q_h^\pi(s, a; \{r_h\}_h) = \mathbb{E}\left[\sum_{h'=h}^H r_{h'}(s_{h'}, a_{h'}) \middle| s_h = s, a_h = a\right], V_h^\pi(s; \{r_h\}_h) = Q_h^\pi(s, \pi_h(s); \{r_h\}_h).$$

For simplicity, we denote $Q_h^\pi(s, a; r) = Q_h^\pi(s, a; \{r_h\}_h)$ and $V_h^\pi(s; r) = V_h^\pi(s; \{r_h\}_h)$. We define the optimal value function $\{V_h^*\}_{h=1}^H$ and the optimal action-value function $\{Q_h^*\}_{h=1}^H$ as $V_h^*(s; r) = \sup_\pi V_h^\pi(s; r)$ and $Q_h^*(s, a; r) = \sup_\pi Q_h^\pi(s, a; r)$ respectively. For any function $V : \mathcal{S} \to \mathbb{R}$, we denote $[\mathbb{P}V](s, a; r) = \mathbb{E}_{s' \sim \mathbb{P}(\cdot|s,a)} V(s'; r)$, and denote the variance of $V$ as

$$[\mathbb{V}f](s, a) = [\mathbb{P}f^2](s, a) - \left([\mathbb{P}f](s, a)^2\right). \tag{3.1}$$

In particular, we have the following Bellman equation, as well as the Bellman optimality equation:

$$Q_h^\pi(s, a; r) = r_h(s, a) + [\mathbb{P}V_{h+1}^\pi](s, a; r), Q_h^*(s, a; r) = r_h(s, a) + [\mathbb{P}V_{h+1}^*](s, a; r).$$

In this paper, we focus on *model-based* algorithms and consider the following *linear mixture/kernel MDP* [16, 7, 3, 30], which assumes that the transition probability $\mathbb{P}$ is a linear mixture of $d$ signed basis measures. Meanwhile, for any function $V$, we assume that we can do the summation $\sum_{s' \in \mathcal{S}} \phi(s'|s, a)V(s)$ efficiently, e.g., using Monte Carlo method [22].

**Definition 3.1** (Linear Mixture MDPs [7, 3, 30])**.** The unknown transition probability $\mathbb{P}$ is a linear combination of $d$ signed basis measures $\phi_i(s'|s, a)$, i.e., $\mathbb{P}(s'|s, a) = \sum_{i=1}^d \phi_i(s'|s, a)\theta_i^*$. Meanwhile, for any $V : \mathcal{S} \to [0, 1], i \in [d], (s, a) \in \mathcal{S} \times \mathcal{A}$, the summation $\sum_{s' \in \mathcal{S}} \phi_i(s'|s, a)V(s')$ is computable. For simplicity, let $\phi = [\phi_1, \ldots, \phi_d]^\top$, $\theta^* = [\theta_1^*, \ldots, \theta_d^*]^\top$ and $\psi_V(s, a) = \sum_{s' \in \mathcal{S}} \phi(s'|s, a)V(s)$. Without loss of generality, we assume $\|\theta^*\|_2 \leq B, \|\psi_V(s, a)\|_2 \leq 1$ for all $V : \mathcal{S} \to [0, 1]$ and $(s, a) \in \mathcal{S} \times \mathcal{A}$.

**Remark 3.2.** A similar but notably different definition (i.e., linear MDPs [21, 10]) has been used in [18], which assumes that $\mathbb{P}(s'|s, a) = \langle \phi(s, a), \mu(s') \rangle$ and $r_h = \langle \phi(s, a), \theta_h \rangle$, $\mu_h(\cdot)$ is a measure and $\theta_h$ is an unknown vector. Comparing with linear MDPs, linear mixture MDPs do not need the reward function $r$ to be linear, which makes our algorithms more general.

With Definition 3.1, it is easy to verify that the expectation of any bounded function $V$ is a linear function of $\psi$:

$$[\mathbb{P}V](s, a) = \langle \psi_V(s, a), \theta^* \rangle. \tag{3.2}$$

**Reward-free RL** For reward-free RL, the algorithm can be divided into two phases: *exploration phase* and *planning phase*. In the exploration phase, the algorithm cannot access the reward function but collect $K$ episodes by doing exploration. In the planning phase, the algorithm is given a series of reward functions and find the optimal policy based on these reward functions, using the $K$ episodes collected in the exploration phase. We formally define $(\epsilon, \delta)$-learn and sample complexity of the algorithm as follows [9].

**Definition 3.3** ($(\epsilon, \delta)$-learnability)**.** Given an MDP transition kernel set $\mathcal{P}$, reward function set $\mathcal{R}$ and a initial state distribution $\mu$, we say a reward-free algorithm can $(\epsilon, \delta)$-learn the problem $(\mathcal{P}, \mathcal{R})$ with sample complexity $K(\epsilon, \delta)$, if for any transition kernel $P \in \mathcal{P}$, after receiving $K(\epsilon, \delta)$ episodes in the exploration phase, for any reward function $r \in \mathcal{R}$, the algorithm returns a policy $\pi$ in planning phase, such that with probability at least $1 - \delta$, $\mathbb{E}_{s_1 \sim \mu}[V_1^*(s_1; r) - V_1^\pi(s_1; r)] \leq \epsilon$.

## 4 Algorithm and Main Results

In this section, we propose a reward-free algorithm. This algorithm works as follows: Firstly, during the *exploration phase*, it samples the MDP episodes, build an estimator $\theta$ for the MDP parameter $\theta^*$, and compute the covariance matrix $\Sigma$ of the feature mappings, which characterizes the uncertainty of the estimator $\theta$. Secondly, during the *planning phase*, the algorithm uses the collected $\theta$ and $\Sigma$ in the exploration phase to find the optimal policy $\pi$ based on the given reward functions.

### 4.1 Planning phase algorithm

We first introduce the PLAN function (Algorithm 1), which is a common module in both planning phase and exploration phase. Given a series of reward functions $\{r_h\}_h$, the goal of PLAN function

---

**Algorithm 1** UCRL-RFE Planning Module (PLAN)

---

**Input:** Estimated parameter and covariance $\boldsymbol{\theta}, \boldsymbol{\Sigma}$, reward $\{r_h\}_{h=1}^H$, parameter $\beta$.
1: For consistency, set $Q_{H+1}(\cdot, \cdot) \leftarrow V_{H+1}(\cdot) \leftarrow 0$
2: **for** $h = H, H-1, \cdots, 1$ **do**
3:      Compute Q function as $Q_h(\cdot, \cdot) \leftarrow \left[ r_h(\cdot, \cdot) + \langle \boldsymbol{\psi}_{V_{h+1}}(\cdot, \cdot), \boldsymbol{\theta} \rangle + \beta \| \boldsymbol{\psi}_{V_{h+1}}(\cdot, \cdot) \|_{\boldsymbol{\Sigma}^{-1}} \right]_{(0, H)}$
4:      Compute value function $V_h(\cdot) \leftarrow \max_{a \in \mathcal{A}} Q_h(\cdot, a)$
5:      Compute policy as $\pi_h(\cdot) \leftarrow \mathrm{argmax}_{a \in \mathcal{A}} Q_h(\cdot, a)$.
6: **end for**
**Output:** Policy $\pi \leftarrow \{\pi_h\}_{h=1}^H$ and $\{V_h\}_{h=1}^H$

---

is to output the optimal policies $\{\pi_h\}_h$ and Q-functions $\{Q_h\}_h$ corresponding to $\{r_h\}_h$. Suppose the unknown parameter $\boldsymbol{\theta}^*$ is known, we can compute $\{Q_h\}_h$ recursively by the following Bellman equation:

$$Q_h(s, a; r) = r_h(s, a) + [\mathbb{P}V_{h+1}](s, a; r) = r_h(s, a) + \langle \boldsymbol{\psi}_{V_{h+1}}(s, a), \boldsymbol{\theta}^* \rangle. \tag{4.1}$$

However, since $\boldsymbol{\theta}^*$ is unknown, we cannot compute $Q_h$ as in (4.1). Instead, PLAN takes the estimated parameter $\boldsymbol{\theta}$ and the "covariance matrix" $\boldsymbol{\Sigma}$ as input. To calculate $Q_h$, PLAN replaces $\boldsymbol{\theta}^*$ with the estimated $\boldsymbol{\theta}$ and plus an additional exploration bonus term $\beta \| \boldsymbol{\psi}_{V_{h+1}}(\cdot, \cdot) \|_{\boldsymbol{\Sigma}^{-1}}$ to (4.1), as in Line 3 of Algorithm 1. Then PLAN takes the greedy policy of the calculated optimistic $Q_h$ and proceeds to the previous step. Finally, the algorithm returns policy $\pi$ in Line 5 as well as the estimated value functions $\{V_h\}_h$.

### 4.2 Exploration phase algorithm

Based on the introduced PLAN function, we propose the UCRL-RFE algorithm in Algorithm 2. In general, UCRL-RFE guides the agent to explore the unknown state space without the information of the reward functions. In detail, for the $k$-th episode, UCRL-RFE first defines the *exploration driven reward function* as follows:

$$r_h^k(s, a) = \min \left\{ 1, \frac{2\beta}{H} \sqrt{\max_{f \in \mathcal{S} \mapsto [0, H-h]} \| \boldsymbol{\psi}_f(s, a) \|_{\boldsymbol{\Sigma}_{1,k}^{-1}}} \right\}, \tag{4.2}$$

where $\boldsymbol{\Sigma}_{1,k}$ is the "covariance matrix" of the feature mapping. Intuitively speaking, $r_h^k(s, a)$ represents the maximum possible uncertainty level of the state-action pair $(s, a)$ caused by the randomness of the MDP transition function, which is *independent* of the true reward functions. Therefore, in order to obtain a good estimation of the optimal policy for any *given* reward functions, it suffices to obtain the optimal policy for $r_h^k(s, a)$. Thus, after obtaining $\{r_h^k\}_h$, UCRL-RFE finds the corresponding near-optimal policies $\{\pi_h^k\}_h$ using PLAN function, with the estimated parameter $\boldsymbol{\theta}_k$ and the "covariance matrix" $\boldsymbol{\Sigma}_{1,k}$ as input. UCRL-RFE uses $\{\pi_h^k\}_h$ as its exploration policy and observes the new episode $s_1^k, a_1^k, \ldots, s_H^k, a_H^k$ induced by $\{\pi_h^k\}_h$.

Next, UCRL-RFE needs to compute the parameters $\boldsymbol{\theta}_{k+1}$ and $\boldsymbol{\Sigma}_{1,k+1}$ for planning in the next episode. Similar to UCRL-VTR proposed by [7, 3], UCRL-RFE also uses a "value-targeted regression (VTR)" estimator, which computes $\boldsymbol{\theta}_{k+1}$ as the minimizer to a ridge regression problem with the target being the past value functions. The main difference between UCRL-RFE and UCRL-VTR is that, due to the lack of true reward functions, UCRL-RFE can not use the estimated value functions as its regression targets. Instead, UCRL-RFE defines the following *pseudo value function* $u_h^k$:

$$u_h^k = \mathrm{argmax}_{f \in \mathcal{S} \mapsto [0, H-h]} \boldsymbol{\psi}_f^\top(s_h^k, a_h^k) \boldsymbol{\Sigma}_{1,k}^{-1} \boldsymbol{\psi}_f(s_h^k, a_h^k). \tag{4.3}$$

Here, $u_h^k$ maximizes the "uncertainty" caused by the transition kernel, which will help the agent to explore the state space. Now given the pseudo value functions, Algorithm 2 computes the estimated $\boldsymbol{\theta}_{k+1}$ as the minimizer to the following ridge regression problem:

$$\boldsymbol{\theta}_{k+1} \leftarrow \mathrm{argmin}_{\boldsymbol{\theta}} \lambda \|\boldsymbol{\theta}\|_2^2 + \sum_{k'=1}^k \sum_{h=1}^H \left( \langle \boldsymbol{\theta}, \boldsymbol{\psi}_{u_h^{k'}}(s_h^{k'}, a_h^{k'}) \rangle - u_h^{k'}(s_{h+1}^{k'}) \right)^2, \tag{4.4}$$

which has a closed-form solution as in Line 12. It also updates the covariance matrix $\boldsymbol{\Sigma}_{1,k+1}$ as in Line 12, by the observed feature mapping $\{\boldsymbol{\psi}_{u_h^k}(s_h^k, a_h^k)\}_h$ in the current episode. In the end, after collecting $HK$ state-action samples, UCRL-RFE calculates the policy $\{\pi_h\}$ as output based on $\boldsymbol{\theta}_{K+1}$ and $\boldsymbol{\Sigma}_{1,K+1}$.

---

**Algorithm 2** UCRL-RFE (Hoeffding Bonus)

---
**Input:** Confident parameter $\beta$, regularization parameter $\lambda$
 1: **Phase I: Exploration Phase**
 2:  Initialize $\boldsymbol{\Sigma}_{1,1} \leftarrow \lambda\mathbf{I}, \mathbf{b}_1 \leftarrow \boldsymbol{\theta}_1 \leftarrow \mathbf{0}$
 3: **for** $k = 1, 2, \cdots, K$ **do**
 4:   Compute the exploration driven reward function $\{r_h^k(\cdot, \cdot)\}_{h=1}^H$ according to (4.2)
 5:   Compute exploration policy and value function as $(\{\pi_h^k\}_{h=1}^H, \{V_h^k\}_{h=1}^H) \leftarrow \texttt{PLAN}(\boldsymbol{\theta}_k, \boldsymbol{\Sigma}_{1,k}, \{r_h^k\}_{h=1}^H, \beta)$
 6:   Receive the initial state $s_1^k \sim \mu$
 7:   **for** $h = 1, 2, \cdots, H$ **do**
 8:     Take action $a_h^k \leftarrow \pi_h^k(s_h^k)$ and receive $s_{h+1}^k$
 9:     Calculate $u_h^k$ for $s_h^k, a_h^k$ according to (4.3)
10:     Set $\boldsymbol{\Sigma}_{h+1,k} \leftarrow \boldsymbol{\Sigma}_{h,k} + \boldsymbol{\psi}_{u_h^k}(s_h^k, a_h^k)\boldsymbol{\psi}_{u_h^k}(s_h^k, a_h^k)^\top, \mathbf{b}_{h+1,k} \leftarrow \mathbf{b}_{h,k} + \boldsymbol{\psi}_{u_h^k}(s_h^k, a_h^k)u_h^k(s_{h+1}^k)$
11:   **end for**
12:   Set $\boldsymbol{\Sigma}_{1,k+1} \leftarrow \boldsymbol{\Sigma}_{H+1,k}, \mathbf{b}_{1,k+1} \leftarrow \mathbf{b}_{H+1,k}, \boldsymbol{\theta}_{k+1} \leftarrow \boldsymbol{\Sigma}_{1,k+1}^{-1}\mathbf{b}_{1,k+1}$
13: **end for**
14: **Phase II: Planning Phase**
15: Receive target reward function $\{r_h\}_{h=1}^H$
16: Compute policy as $(\{\pi_h\}_{h=1}^H, \{V_h\}_{h=1}^H) \leftarrow \texttt{PLAN}(\boldsymbol{\theta}_{K+1}, \boldsymbol{\Sigma}_{1,K+1}, \{r_h\}_{h=1}^H, \beta)$
**Output:** Policy $\{\pi_h\}_{h=1}^H$

---

**Remark 4.1.** Here we do a comparison between our UCRL-RFE and the reward-free RL algorithm in [18]. The main difference is that Wang et al. [18] estimates $\boldsymbol{\theta}_k$ by regression with value function $V_h^k$ being the target, while our UCRL-RFE does regression with the pseudo value function $u_h^k$ being the target. That is mainly due to the different problem settings (linear MDP v.s. linear mixture MDP).

### 4.3 Implementation details

In general, solving the maximization problem (4.3) is hard. Here, we provide a simple approximate solution to the problem (4.2) and (4.3) for the finite state space case ($|\mathcal{S}| < \infty$). Instead of maximizing the $\ell_2$ norm-based objective $\left\|\boldsymbol{\Sigma}_{1,k}^{-1/2}\boldsymbol{\psi}_f(s_h^k, a_h^k)\right\|_2$, we write $\boldsymbol{\psi}_f(s, a) = \boldsymbol{\Phi}(s, a)\mathbf{f}$ with $\boldsymbol{\Phi}(s, a) = (\boldsymbol{\phi}(s, a, S_1), \cdots, \boldsymbol{\phi}(s, a, S_{|\mathcal{S}|}))$ and $\mathbf{f} = (f(S_1), \cdots, f(S_{|\mathcal{S}|}))^\top$, relax the $\ell_2$ norm into $\ell_1$ norm since $\|\mathbf{x}\|_2 \geq \|\mathbf{x}_1\|_1/\sqrt{d}$ for any $\mathbf{x} \in \mathbb{R}^d$, and maximize the following $\ell_1$ norm-based objective

$$\max_{\mathbf{f}} \left\|\boldsymbol{\Sigma}_{1,k}^{-1/2}\boldsymbol{\Phi}(s, a)\mathbf{f}\right\|_1 \text{ subject to } \|\mathbf{f}\|_\infty \leq H - h. \tag{4.5}$$

(4.5) can be formulated as a linear programming, which can be solved by interior method [11] or simplex method [5] efficiently. Since $\|\mathbf{x}\|_1/\sqrt{d} \leq \|\mathbf{x}\|_2 \leq \|\mathbf{x}\|_1$, the performance of this approximate solution is guaranteed. For the case where the state space is infinite, we can use state aggregation methods such as soft state aggregation [15] to reduce the infinite state space to finite state space and then apply the above approximate solution to solve it.

### 4.4 Sample complexity

Now we provide the sample complexity for Algorithm 2.

**Theorem 4.2** (Sample complexity of UCRL-RFE). For Algorithm 2, setting parameter $\beta = H\sqrt{d\log(3(1 + KH^3B^2)/\delta)} + 1$, $\lambda = B^{-2}$, then for any $0 < \epsilon < 1$, if $K = \widetilde{\mathcal{O}}(H^5d^2\epsilon^{-2})$, we have with probability at least $1 - \delta$ that, $\mathbb{E}_{s\sim\mu}[V_1^*(s; r) - V_1^\pi(s; r)] \leq \epsilon$.

**Remark 4.3.** Theorem 4.2 shows that UCRL-RFE only needs $\text{poly}(d, H, \epsilon^{-1})$ sample complexity to find an $\epsilon$-optimal policy, which suggests that model-based reward-free algorithm is sample-efficient. Thanks to linear function approximation, the sample complexity only depends on the dimension of the feature mapping $d$ and the length of the episode and does not depend on the cardinalities of the state and action spaces.

**Corollary 4.4.** Under the same conditions as in Theorem 4.2, if solving the relaxed optimization problem in (4.5), Algorithm 2 has $K = \widetilde{\mathcal{O}}(H^5 d^3 \epsilon^{-2})$ sample complexity.

# 5 Improved Algorithm with Bernstein Bonus

Theorem 4.2 suggests that UCRL-RFE in Algorithm 2 enjoys an $\widetilde{\mathcal{O}}(H^5 d^2 \epsilon^{-2})$ sample complexity to find an $\epsilon$-optimal policy. In this section, we seek to further improve the sample complexity.

A key observation is that for any given reward functions $\{r_h\}_h$, the error between the exploration policy $\{\pi_h\}_h$ and the optimal policy can be decomposed into two parts: *the exploration error* which is the difference between $\{r_h\}_h$ and the exploration driven reward function $\{r_h^k\}_h$, and *the approximation error* which is the difference between the optimal value function $V_1^*(\cdot; r_h^k)$ and our estimated value function $V_1^{\pi_h^k}(\cdot; r_h^k)$ with respect to $\{r_h^k\}_h$. For the latter one, our exploration strategy adapted from VTR is often too conservative since it does not distinguish different value functions and state-action pairs from different episodes and steps. Therefore, inspired by [29], we propose a variant of UCRL-RFE called UCRL-RFE+, which adopts a Bernstein-type bonus for exploration and achieves a better sample complexity.

---

**Algorithm 3** UCRL-RFE+ (Bernstein Bonus)

---

**Input:** Parameter $\beta, \widehat{\beta}, \widetilde{\beta}, \check{\beta}$, regularization parameter $\lambda$
 1: **Stage I: Exploration Phase**
 2: Initialize $\boldsymbol{\Sigma}_{1,1} = \widehat{\boldsymbol{\Sigma}}_{1,1} = \widetilde{\boldsymbol{\Sigma}}_{1,1} = \lambda \mathbf{I}, \mathbf{b}_1 = \widehat{\mathbf{b}}_1 = \widetilde{\mathbf{b}}_1 = \boldsymbol{\theta}_1 = \widehat{\boldsymbol{\theta}}_1 = \widetilde{\boldsymbol{\theta}}_1 = \mathbf{0}$
 3: **for** $k = 1, 2, \cdots, K$ **do**
 4:     Set $\{r_h^k(\cdot, \cdot)\}_{h=1}^H$ to (4.2).
 5:     Compute exploration policy and value function as $(\{\pi_h^k\}_{h=1}^H, \{V_h^k\}_{h=1}^H) \leftarrow \mathtt{PLAN}(\widehat{\boldsymbol{\theta}}_k, \widehat{\boldsymbol{\Sigma}}_{1,k}, \{r_h^k\}_{h=1}^H, \widehat{\beta})$
 6:     Receive the initial state $s_1^k \sim \mu$.
 7:     **for** $h = 1, 2, \cdots, H$ **do**
 8:         Take action $a_h^k = \pi_h^k(s_h^k)$ and receive $s_{h+1}^k$
 9:         Calculate $u_h^k, \nu_h^k$ for $s_h^k, a_h^k$ according to (4.3) and (5.2) separately
10:         Set $\boldsymbol{\Sigma}_{h+1,k} \leftarrow \boldsymbol{\Sigma}_{h,k} + \boldsymbol{\psi}_{u_h^k}(s_h^k, a_h^k)\boldsymbol{\psi}_{u_h^k}(s_h^k, a_h^k)^\top$
11:         Set $\widehat{\boldsymbol{\Sigma}}_{h+1,k}, \widetilde{\boldsymbol{\Sigma}}_{h+1,k}, \widehat{\mathbf{b}}_{h+1,k}, \widetilde{\mathbf{b}}_{h+1,k}$ using (5.4)
12:     **end for**
13:     Set $\boldsymbol{\Sigma}_{1,k+1} \leftarrow \boldsymbol{\Sigma}_{H+1,k}$
14:     Set $\widehat{\boldsymbol{\Sigma}}_{1,k+1} \leftarrow \widehat{\boldsymbol{\Sigma}}_{H+1,k}, \widehat{\mathbf{b}}_{1,k+1} \leftarrow \widehat{\mathbf{b}}_{H+1,k}, \widehat{\boldsymbol{\theta}}_{k+1} \leftarrow \widehat{\boldsymbol{\Sigma}}_{1,k+1}^{-1}\widehat{\mathbf{b}}_{1,k+1}$
15:     Set $\widetilde{\boldsymbol{\Sigma}}_{1,k+1} \leftarrow \widetilde{\boldsymbol{\Sigma}}_{H+1,k}, \widetilde{\mathbf{b}}_{1,k+1} \leftarrow \widetilde{\mathbf{b}}_{H+1,k}, \widetilde{\boldsymbol{\theta}}_{k+1} \leftarrow \widetilde{\boldsymbol{\Sigma}}_{1,k+1}^{-1}\widetilde{\mathbf{b}}_{1,k+1}$
16: **end for**
17: Set $\boldsymbol{\theta}_{K+1} \leftarrow \boldsymbol{\Sigma}_{1,K+1}^{-1} \sum_{k=1}^K \sum_{h=1}^H \boldsymbol{\psi}_{u_h^k}(s_h^k, a_h^k) u_h^k(s_{h+1}^k)$
18: **Stage II: Planning Phase**
19: Receive target reward function $\{r_h\}_{h=1}^H$
20: Compute exploration policy as $(\{\pi_h\}_{h=1}^H, \{V_h\}_{h=1}^H) \leftarrow \mathtt{PLAN}(\boldsymbol{\theta}_{K+1}, \boldsymbol{\Sigma}_{1,K+1}, \{r_h\}_{h=1}^H, \beta)$
**Output:** Policy $\{\pi_h\}_{h=1}^H$

---

## 5.1 Exploration phase algorithm with Bernstein bonus

UCRL-RFE+ in presented in Algorithm 3. The algorithm structure is similar to that of UCRL-RFE, which can be decomposed into the exploration phase and planning phase. There are two main differences. First, in contrast to UCRL-RFE which uses $\boldsymbol{\theta}_k$ for the $\mathtt{PLAN}$ function in both exploration and planning phases, UCRL-RFE+ only uses $\boldsymbol{\theta}_{K+1}$ for the $\mathtt{PLAN}$ function in the planning phase. For the exploration phase, UCRL-RFE+ constructs a new estimator $\widehat{\boldsymbol{\theta}}_k$ based on $\{V_{h+1}^{k'}\}_{k' \leq k-1, h}$, which are the value functions of the exploration driven rewards. Second, to build $\widehat{\boldsymbol{\theta}}_k$, one way is to choose it as the solution to the ridge regression problem with contexts $\boldsymbol{\psi}_{V_{h+1}^{k'}}(s_h^{k'}, a_h^{k'})$ and targets $V_{h+1}^{k'}(s_{h+1}^{k'})$, similar to (4.4). However, since the targets $V_{h+1}^{k'}(s_{h+1}^{k'})$ have different variances at different steps and episodes, we are actually facing a *heteroscedastic linear regression* problem. Therefore, inspired by a recent line of work [29, 20] which use Bernstein inequality for vector-valued self-normalized

martingale to construct a tighter confidence ball for exploration, we also incorporate the variance to build choose $\widehat{\boldsymbol{\theta}}_k$ as the solution to the following *weighted ridge regression* problem, which is an enhanced estimator for the heteroscedastic case:

$$\widehat{\boldsymbol{\theta}}_k \leftarrow \operatorname*{argmin}_{\boldsymbol{\theta}} \lambda \|\boldsymbol{\theta}\|_2^2 + \sum_{k'=1}^{k-1} \sum_{h=1}^{H} \left( \langle \boldsymbol{\theta}, \boldsymbol{\psi}_{V_{h+1}^{k'}}(s_h^{k'}, a_h^{k'}) \rangle - V_{h+1}^{k'}(s_{h+1}^{k'}) \right)^2 / [\sigma_h^{k'}]^2, \qquad (5.1)$$

where $[\sigma_h^{k'}]^2$ is the variance of $V_{h+1}^{k'}(s_{h+1}^{k'})$. The idea to use variances to improve the sample complexity is closely related to the use of "Bernstein bonus" in reward-free RL for the tabular MDPs [12, 28, 14]. Since $\sigma_h^{k'}$ is unknown, we will use $\nu_h^{k'} = [\bar{\sigma}_h^{k'}]^2$ as a plug-in estimator to replace $[\sigma_h^{k'}]^2$ in (5.1). After obtaining $\widehat{\boldsymbol{\theta}}_k$, UCRL-RFE+ sets the $\widehat{\boldsymbol{\Sigma}}_{1,k}$ as the covariance matrix of the features $\boldsymbol{\psi}_{V_{h+1}^k}(s_h^k, a_h^k)/\bar{\sigma}_h^k$, and feeds it into the PLAN function with the exploration-driven reward functions and the confidence radius $\widehat{\beta}$. UCRL-RFE+ takes the output $\{\pi_h^k\}_h$ as the exploration policy, and $\{V_h^k\}_h$ as the value functions to construct the estimator $\widehat{\boldsymbol{\theta}}_{k+1}$ for next episode. In the end, when it comes to the planning phase, after receiving reward functions $\{r_h\}_h$, UCRL-RFE+ takes $\boldsymbol{\theta}_{K+1}$ as the solution to the ridge regression problem with contexts $\{\boldsymbol{\psi}_{u_h^k}(s_h^k, a_h^k)\}_{k,h}$ and targets $\{u_h^k(s_{h+1}^k)\}_{k,h}$, and the covariance matrix $\boldsymbol{\Sigma}_{1,K+1}$ as input, and uses PLAN to find the near optimal policy $\{\pi_h\}_h$ with confidence radius $\beta$. It remains to specify $\nu_h^k$ in the weighted ridge regression. On the one hand, we need $\nu_h^k$ to be an upper bound of $[\sigma_h^k]^2$. On the other hand, we require $\nu_h^k$ to have a strictly positive lower bound to let (5.1) be valid. Therefore, we construct $\nu_h^k$ as follows:

$$\nu_h^k = \max\{\alpha, \bar{\mathbb{V}}_h^k(s_h^k, a_h^k) + E_k^h(s_h^k, a_h^k)\}, \qquad (5.2)$$

where $\bar{\mathbb{V}}_h^k$ is the estimated variance of value function $V_h^k$ and $E_h^k$ is a correction term to calibrate the estimated variance, and $\alpha > 0$ is a positive constant. To compute $\bar{\mathbb{V}}_h^k(s_h^k, a_h^k)$, considering the following fact:

$$[\mathbb{V}V_{h+1}^k](s,a) = [\mathbb{P}[V_{h+1}^k]^2](s,a) - [\mathbb{P}V_{h+1}^k](s,a)^2 = \langle \boldsymbol{\theta}^*, \boldsymbol{\psi}_{[V_{h+1}^k]^2}(s,a) \rangle - \langle \boldsymbol{\theta}^*, \boldsymbol{\psi}_{V_{h+1}^k}(s,a) \rangle^2,$$

it suffices to estimate $\langle \boldsymbol{\theta}^*, \boldsymbol{\psi}_{[V_{h+1}^k]^2}(s,a) \rangle$ and $\langle \boldsymbol{\theta}^*, \boldsymbol{\psi}_{V_{h+1}^k}(s,a) \rangle$ separately. For the first term, $\boldsymbol{\theta}^*$ can be regarded as the unknown parameter of a regression problem w.r.t. contexts $\boldsymbol{\psi}_{[V_{h+1}^k]^2}(s_h^{k'}, a_h^{k'})$ and targets $\boldsymbol{\psi}_{[V_{h+1}^{k'}]^2}(s_h^{k'}, a_h^{k'})$. Therefore, the first term can be estimated by $\langle \boldsymbol{\psi}_{[V_{h+1}^k]^2}(s,a), \widetilde{\boldsymbol{\theta}}_k \rangle$, where

$$\widetilde{\boldsymbol{\theta}}_k \leftarrow \operatorname*{argmin}_{\boldsymbol{\theta}} \lambda \|\boldsymbol{\theta}\|_2^2 + \sum_{k'=1}^{k-1} \sum_{h=1}^{H} \left( \langle \boldsymbol{\theta}, \boldsymbol{\psi}_{[V_{h+1}^{k'}]^2}(s_h^{k'}, a_h^{k'}) \rangle - [V_{h+1}^{k'}(s_{h+1}^{k'})]^2 \right)^2.$$

In addition, the second term $\langle \boldsymbol{\theta}^*, \boldsymbol{\psi}_{V_{h+1}^k}(s,a) \rangle$ can be approximated by $\langle \boldsymbol{\psi}_{V_{h+1}^k}(s,a), \widehat{\boldsymbol{\theta}}_k \rangle$. Therefore, the final estimator $[\bar{\mathbb{V}}V_{h+1}^k](s,a)$ is defined as

$$\bar{\mathbb{V}}_h^k(s,a) = \left[ \langle \boldsymbol{\psi}_{[V_{h+1}^k]^2}(s,a), \widetilde{\boldsymbol{\theta}}_k \rangle \right]_{(0,H^2)} - \left[ \langle \boldsymbol{\psi}_{V_{h+1}^k}(s,a), \widehat{\boldsymbol{\theta}}_k \rangle \right]_{(0,H)}^2. \qquad (5.3)$$

For the correction terms $E_h^k$, we define it as follows:

$$E_h^k(s,a) = \min \left\{ H^2, \widetilde{\beta} \big\| \boldsymbol{\psi}_{[V_{h+1}^k]^2}(s,a) \big\|_{\widetilde{\boldsymbol{\Sigma}}_{1,k}^{-1}} \right\} + \min \left\{ H^2, 2H\breve{\beta} \big\| \boldsymbol{\psi}_{V_{h+1}^k}(s,a) \big\|_{\widehat{\boldsymbol{\Sigma}}_{1,k}^{-1}} \right\},$$

where $\widetilde{\boldsymbol{\Sigma}}_{1,k}$ is the covariance matrix of the features $\boldsymbol{\psi}_{[V_{h+1}^{k'}]^2}(s_h^{k'}, a_h^{k'})$, $\widetilde{\beta}, \breve{\beta}$ are two confidence radius. It can be shown that, with these definitions, $\bar{\mathbb{V}}_h^k(s,a) + E_h^k(s,a)$ is an upper bound of $[\sigma_h^k]^2$.

Finally, to enable online update, UCRL-RFE+ updates its covariance matrices recursively as follows, along with sequences $\widehat{\mathbf{b}}_h^k, \widetilde{\mathbf{b}}_h^k$:

$$\widehat{\boldsymbol{\Sigma}}_{h+1,k} \leftarrow \widehat{\boldsymbol{\Sigma}}_{h,k} + \boldsymbol{\psi}_{V_{h+1}^k}(s_h^k, a_h^k)\boldsymbol{\psi}_{V_{h+1}^k}(s_h^k, a_h^k)^\top / \nu_h^k$$

$$\widetilde{\boldsymbol{\Sigma}}_{h+1,k} \leftarrow \widetilde{\boldsymbol{\Sigma}}_{h,k} + \boldsymbol{\psi}_{[V_{h+1}^k]^2}(s_h^k, a_h^k)\boldsymbol{\psi}_{[V_{h+1}^k]^2}(s_h^k, a_h^k)^\top$$

$$\widehat{\mathbf{b}}_{h+1,k} \leftarrow \widehat{\mathbf{b}}_{h,k} + \boldsymbol{\psi}_{V_{h+1}^k}(s_h^k, a_h^k)V_{h+1}^k(s_{h+1}^k) / \nu_h^k$$

$$\widetilde{\mathbf{b}}_{h+1,k} \leftarrow \widetilde{\mathbf{b}}_{h,k} + \boldsymbol{\psi}_{[V_{h+1}^k]^2}(s_h^k, a_h^k)[V_{h+1}^k(s_{h+1}^k)]^2, \qquad (5.4)$$

where $u_h^k$ is the pseudo value function in (4.3) and $\nu_h^k$ is defined in (5.2). Then UCRL-RFE+ computes $\widehat{\boldsymbol{\theta}}_k, \widetilde{\boldsymbol{\theta}}_k$ as in Line 14 to Line 15 of Algorithm 3.

## 5.2 Sample complexity

Now we present the sample complexity for Algorithm 3.

**Theorem 5.1** (Sample complexity of UCRL-RFE+). For Algorithm 3, setting $\lambda = B^{-2}, \alpha = H^2/d$ in (5.2), and the confidence radius as

$$\widehat{\beta} = 8\sqrt{d\log(1 + KHB^2)\log(48K^2H^2/\delta)} + 4\sqrt{d}\log(48K^2H^2/\delta) + 1$$
$$\check{\beta} = 8d\sqrt{\log(1 + KHB^2)\log(48K^2H^2/\delta)} + 4\sqrt{d}\log(48K^2H^2/\delta) + 1$$
$$\widetilde{\beta} = 8H^2\sqrt{d\log(1 + KHB^2)\log(48K^2H^2/\delta)} + 4H^2\log(48K^2H^2/\delta) + 1$$
$$\beta = H\sqrt{d\log(12(1 + KH^3B^2)/\delta)} + 1,$$

then for any $0 < \epsilon < 1$, if $K = \widetilde{\mathcal{O}}(H^4d(H + d)\epsilon^{-2})$, then with probability at least $1 - \delta$, we have $\mathbb{E}_{s\sim\mu}[V_1^*(s; r) - V_1^\pi(s; r)] \leq \epsilon$.

**Remark 5.2.** Theorem 5.1 suggests that when $d \geq H$, the sample complexity of UCRL-RFE+ is $\widetilde{\mathcal{O}}(H^4d^2\epsilon^{-2})$, which improves the sample complexity of UCRL-RFE by a factor of $H$. On the other hand, when $H \geq d$, the sample complexity of UCRL-RFE+ reduces to $\widetilde{\mathcal{O}}(H^5d\epsilon^{-2})$, which is better than that of UCRL-RFE by a factor of $d$. At a high-level, the sample complexity improvement is attributed to the Bernstein-type bonus.

**Corollary 5.3.** Under the same conditions as in Theorem 5.1, if solving the relaxed optimization problem in (4.5), Algorithm 3 has $K = \widetilde{\mathcal{O}}(H^5d^3\epsilon^{-2})$ sample complexity.

## 6 Lower Bound of Sample Complexity

In this section, we will provide a lower bound of sample complexity for reward-free RL under linear mixture MDP setting.

The proof is by construction. Given $d \geq 2$, we first define a binary vector set $\mathcal{M} = \{\mathbf{x}|\mathbf{x} \in \mathbb{R}^{d-1}, [\mathbf{x}]_i \in \{-1, 1\}\}$. We index each vector in $\mathcal{M}$ as $\mathbf{x}_1, \mathbf{x}_2, \cdots, \mathbf{x}_{|\mathcal{M}|}$. Equipped with the set $\mathcal{M}$, we construct a class of MDPs. As shown in Figure 1, there are in total three states $S_1, S_{2,1}, S_{2,2}$ and $|\mathcal{A}| = |\mathcal{M}|$ actions $a_1, a_2, \cdots a_{|\mathcal{A}|}$. We define the feature mapping $\phi(s'|s, a_i) \in \mathbb{R}^d$ as follows:

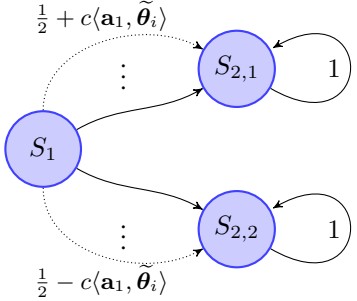

Figure 1: The transition kernel $\mathbb{P}$ of the class of hard-to-learn linear mixture MDPs. The kernel $\mathbb{P}$ is parameterized by $\boldsymbol{\theta}_i = (\sqrt{2}, \alpha\boldsymbol{\theta}_i^\top/\sqrt{d})^\top$ for some small $\alpha$. $c = \alpha/(\sqrt{2}d)$. The learner knows the MDP structure, but does not know the parameter $\boldsymbol{\theta}_i$ (or $\widetilde{\boldsymbol{\theta}}_i \in \mathcal{M}$).

$$\phi(S_{2,1}|S_1, a_j) = \left(\frac{\sqrt{2}}{4} \quad \frac{\mathbf{a}_j^\top}{\sqrt{2d}}\right)^\top,$$

$$\phi(S_{2,2}|S_1, a_j) = \left(\frac{\sqrt{2}}{4} \quad -\frac{\mathbf{a}_j^\top}{\sqrt{2d}}\right)^\top,$$

$\phi(S_{2,j}|S_{2,j}, a_i) = (1/\sqrt{2} \quad \mathbf{0}^\top)^\top$ for $j = 1, 2$, and $\phi(s'|s, a) = \mathbf{0}$ for all the remaining cases. Furthermore, we define a $d$-dimensional parameter set $\boldsymbol{\Theta} \subseteq \mathbb{R}^{d+1}$ by $\boldsymbol{\Theta} = \{\boldsymbol{\theta}_i|\boldsymbol{\theta}_i = (\sqrt{2}, \alpha\widetilde{\boldsymbol{\theta}}_i^\top/\sqrt{d})^\top\}$ where $\widetilde{\boldsymbol{\theta}}_i = \mathbf{x}_i \in \mathcal{M}$ and $\alpha$ is a small absolute constant. Therefore, for each parameter $\boldsymbol{\theta}_i$, we define the transition probability of the linear mixture MDP as $\mathbb{P}(\cdot|\cdot, \cdot) = \langle\phi(\cdot|\cdot, \cdot), \boldsymbol{\theta}_i\rangle$. Specifically, the transition between $S_1$ and $\{S_{2,1}, S_{2,2}\}$ is represented as

$$\mathbb{P}_{\boldsymbol{\theta}_i}(S_{2,1}|S_1, a_j) = \frac{1}{2} + \frac{\alpha}{\sqrt{2}d}\langle\widetilde{\boldsymbol{\theta}}_i, \mathbf{a}_j\rangle, \quad \mathbb{P}_{\boldsymbol{\theta}_i}(S_{2,2}|S_1, a_j) = \frac{1}{2} - \frac{\alpha}{\sqrt{2}d}\langle\widetilde{\boldsymbol{\theta}}_i, \mathbf{a}_j\rangle.$$

Meanwhile, we have $S_{2,1}$ and $S_{2,2}$ are both absorbing states. With the constructed hard-to-learn MDP class, we can prove the lower bound of sample complexity as follows:

**Theorem 6.1.** Given dimension $d \geq 50$ and $H \geq 2$, set $\epsilon \leq (H-1)/(2\sqrt{2})$ and $\delta \in (0, 1/2)$, then there exists a class of linear mixture MDPs, such that any reward-free RL algorithm that $(\epsilon, \delta)$-learns the problem $(\mathcal{P}, \mathcal{R})$ where $\mathcal{R} = \{\{r_h\}_{h=1}^H, 0 \leq r_h \leq 1\}$, needs to collect at least $K = C(1-\delta)dH^2\epsilon^{-2}$ episodes during exploration, where $C$ is an absolute constant.

**Remark 6.2.** When $d \leq H$, the sample complexity of UCRL-RFE+ is $\widetilde{\mathcal{O}}(H^5 d\epsilon^{-2})$, which matches the lower bound in terms of both $\epsilon$ and $d$, ignoring the logarithmic terms. When $d > H$, the sample complexity of UCRL-RFE+ is $\widetilde{\mathcal{O}}(H^4 d^2 \epsilon^{-2})$, which matches the lower bound only in terms of $\epsilon$. The factor of $d$ gap between the upper and lower bounds is due to the fact that our upper bound holds for the arbitrary number of actions. Such a gap also appears in best-arm identification in the linear bandits problem (See Eq. (3) in Tao et al. [17] with $N = \mathcal{O}(2^d)$). There is also a factor of $H^2$ gap between the upper and lower bounds, and we leave it as future work to remove this gap.

## 7 Conclusion

We studied model-based reward-free exploration for learning the linear mixture MDPs. We proposed two algorithms, UCRL-RFE, and UCRL-RFE+, which are guaranteed to have polynomial sample complexities in exploration to find a near-optimal policy in the planning phase for any given reward function. To our knowledge, these are the first algorithms and theoretical guarantees for model-based reward-free RL with function approximation. We also give a sample complexity lower bound for any reward-free algorithm to learn linear mixture MDPs. The sample complexity of our algorithm UCRL-RFE+ matches the lower bound in terms of the dependence on accuracy $\epsilon$ and feature dimension $d$ when $H \geq d$.

## Acknowledgments and Disclosure of Funding

We thank the anonymous reviewers for their helpful comments. WZ, DZ and QG are partially supported by the National Science Foundation CAREER Award 1906169, IIS-1904183 and AWS Machine Learning Research Award. The views and conclusions contained in this paper are those of the authors and should not be interpreted as representing any funding agencies.

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
