# A   Proofs of Upper Bounds

In this section, we provide the proofs of sample complexity upper bounds.

## A.1   Proof of Theorem 4.2

We will first introduce a lemma to show that for the planning module Algorithm 1, if it is guaranteed that the estimation $\boldsymbol{\theta}$ is close to the true parameter $\boldsymbol{\theta}^*$, then the estimated value function is optimistic. Also the gap between the optimal value function and the value function of the output policy $\{\pi_h\}_{h=1}^H$ could be controlled by the summation of UCB bonus term.

**Lemma A.1.** Let $\boldsymbol{\theta}, \boldsymbol{\Sigma}, \beta$ be as defined in Algorithm 1. Suppose there exists some event $\boldsymbol{\xi}$ such that $\|\boldsymbol{\theta}^* - \boldsymbol{\theta}\|_{\boldsymbol{\Sigma}} \leq \beta$ on this event. Then on this event, for all $s \in \mathcal{S}$, $V_1(s) \geq V_1^*(s; r)$, where $V_1$ is the output value function for Algorithm 1. We also have that

$$V_1(s) - V_1^\pi(s) \leq \mathbb{E}\left[\sum_{h=1}^H \min\{H, 2\beta\|\boldsymbol{\psi}_{V_{h+1}}(s_h, \pi_h(s_h))\|_{\boldsymbol{\Sigma}^{-1}}\}\Big|s, \pi\right],$$

where the policy $\pi = \{\pi_h\}_{h=1}^H$ is generated by the planning module Algorithm 1 and $V_h$ is the value function calculated on Line 5 in Algorithm 1.

Next we will give the lemmas on how to guarantee the condition of Lemma A.1 and how to utilize the result of that lemma to control the final policy error $V_1^*(s_1; r) - V_1^\pi(s_1; r)$ where the policy $\pi$ is output of the planning phase. We start with Algorithm 2, which uses the Hoeffding bonus.

Firstly, the next lemma shows how to guarantee the condition in Lemma A.1.

**Lemma A.2** (Confidence interval, Hoeffding)**.** For Algorithm 2, let $\lambda, \beta$ be as defined in Theorem 4.2, then with probability at least $1 - \delta/3$, $\|\boldsymbol{\theta}^* - \boldsymbol{\theta}_k\|_{\boldsymbol{\Sigma}_{1,k}} \leq \beta$ for any $k \in [K+1]$.

Secondly, based on the lemma above, we find that the policy error during the planning phase is controlled by a summation of the UCB terms. Since from the intuition, the exploration driven reward function (4.2) is the UCB term divided by $H$, the policy error during the planning phase can be converted to the value function $V_1^k$ in the exploration phase. The next lemma shows that the summation of $V_1^k$ over $K$ iterations is sub-linear to $K$, thus the policy error during the planning phase should be small.

**Lemma A.3** (Summation, Hoeffding)**.** Set the parameters of Algorithm 2 as that of Theorem 4.2. If the condition in Lemma A.2 holds, then with probability at least $1 - \delta/3$, the summation of the value function $V_1^k(s_1^k)$ during the exploration phase is controlled by

$$\sum_{k=1}^K V_1^k(s_1^k) \leq 8\beta\sqrt{HKd\log(1 + KH^3B^2/d)}$$

$$+ 8\beta Hd\log(1 + KH^3B^2) + 2H\sqrt{2HK\log(1/\delta)}.$$

Equipped with these lemmas, we are about to prove Theorem 4.2.

*Proof of Theorem 4.2.* In the following proof, we condition on the events in Lemma A.2 and Lemma A.3 which holds with probability at least $1 - 2\delta/3$ by taking the union bound. Applying Lemma A.1 to the final planning phase, we have

$$V_1^*(s; r) - V_1^\pi(s; r) \leq V_1(s; r) - V_1^\pi(s; r) \leq \underbrace{\mathbb{E}\left[\sum_{h=1}^H \min\{H, 2\beta\|\boldsymbol{\psi}_{V_{h+1}}(s_h, \pi_h(s_h))\|_{\boldsymbol{\Sigma}_{1,K+1}^{-1}}\}\right]}_{I_1},$$

$$(\text{A.1})$$

where the expectation is taken condition on initial state $s$ and policy $\pi$ generated by the planning phase. Since $\boldsymbol{\Sigma}_{1,k} \preceq \boldsymbol{\Sigma}_{1,K+1}$ for all $k \in [K]$, we can guarantee that $\|\boldsymbol{\psi}_{V_{h+1}}(s_h, \pi_h(s_h))\|_{\boldsymbol{\Sigma}_{1,K+1}^{-1}} \leq \|\boldsymbol{\psi}_{V_{h+1}}(s_h, \pi_h(s_h))\|_{\boldsymbol{\Sigma}_{1,k}^{-1}}$. Recall the exploration driven reward function is defined by

$$r_h^k(s, a) = \min\left\{1, \frac{2\beta}{H}\sqrt{\max_{f \in \mathcal{S} \mapsto [0, H-h]}\|\boldsymbol{\psi}_f(s, a)\|_{\boldsymbol{\Sigma}_{1,k}^{-1}}}\right\}, (\text{A.3}) \qquad (\text{A.2})$$

one can easily verify that $\min\{H, 2\beta\|\boldsymbol{\psi}_{V_{h+1}}(s_h, \pi_h(s_h))\|_{\boldsymbol{\Sigma}_{1,k}^{-1}}\} \leq Hr_h^k(s_h, \pi_h(s_h))$. Therefore for any $k \in [K]$ episode, we can bound the term $I_1$ using the value function $V_1^\pi(s; \{r_h^k\}_{h=1}^H)$ of the output policy $\pi$ in the planning phase given the $\{r_h^k\}_{h=1}^H$ as the reward function, i.e.

$$I_1 \leq \mathbb{E}\left[\sum_{h=1}^H Hr_h^k(s_h, \pi_h(s_h))\right] = HV_1^\pi(s; \{r_h^k\}_{h=1}^k). \tag{A.3}$$

Plugging the bound of $I_1$ back into (A.1) then taking the expectation over the initial state distribution $\mu$, we have for any $k \in [K]$,

$$\mathbb{E}_{s \sim \mu}[V_1^*(s; r) - V_1^\pi(s; r)] \leq H\mathbb{E}_{s \sim \mu}[V_1^\pi(s; \{r_h^k\}_{h=1}^k)]$$
$$= H\left(V_1^\pi(s_1^k; \{r_h^k\}_{h=1}^k) - V_1^\pi(s_1^k; \{r_h^k\}_{h=1}^k)\right)$$
$$+ H\mathbb{E}_{s \sim \mu}[V_1^\pi(s; \{r_h^k\}_{h=1}^k)].$$

Hence

$$\mathbb{E}_{s \sim \mu}[V_1^*(s; r) - V_1^\pi(s; r)] \leq \frac{H}{K}\sum_{k=1}^K \left(V_1^\pi(s_1^k; \{r_h^k\}_{h=1}^k) - V_1^\pi(s_1^k; \{r_h^k\}_{h=1}^k)\right.$$
$$\left. + \mathbb{E}_{s \sim \mu}[V_1^\pi(s; \{r_h^k\}_{h=1}^k)]\right). \tag{A.4}$$

Since $V_1^\pi(s; \{r_h^k\}_{h=1}^k) \leq H$ for all $k \in [K], s \in \mathcal{S}$, by Azuma-Hoeffding's inequality, with probability at least $1 - \delta/3$,

$$\sum_{k=1}^K \left(\mathbb{E}_{s \sim \mu}[V_1^\pi(s; \{r_h^k\}_{h=1}^k)] - V_1^\pi(s_1^k; \{r_h^k\}_{h=1}^k)\right) \leq H\sqrt{2K\log(3/\delta)}. \tag{A.5}$$

By plugging (A.5) into (A.4), we have

$$\mathbb{E}_{s \sim \mu}[V_1^*(s; r) - V_1^\pi(s; r)] \leq \frac{H}{K}\sum_{k=1}^K V_1^\pi(s_1^k; \{r_h^k\}_{h=1}^k) + H^2\sqrt{2\log(3/\delta)/K}.$$

Applying Lemma A.1 to the exploration phase, for any $k$-th episode, $V_1^\pi(s_1^k; \{r_h^k\}_{h=1}^k) \leq V_1^*(s_1^k; \{r_h^k\}_{h=1}^k) \leq V_1^k(s_1^k)$, thus replacing the value function $V_1^\pi$ with the estimated value function $V_1^k$, we have

$$\mathbb{E}_{s \sim \mu}[V_1^*(s; r) - V_1^\pi(s; r)] \leq \frac{H}{K}\sum_{k=1}^K V_1^k(s_1^k) + H^2\sqrt{2\log(3/\delta)/K}. \tag{A.6}$$

Finally by Lemma A.3 we can bound the summation over $V_1^k$, hence

$$\mathbb{E}_{s \sim \mu}[V_1^*(s; r) - V_1^\pi(s; r)] \leq H^2\sqrt{2\log(3/\delta)/K} + 8\beta\sqrt{H^3 d\log(1 + KH^3B^2/d)/K}$$
$$+ 8\beta dH^2\log(1 + KH^3B^2)/K + 2H^2\sqrt{2H\log(1/\delta)/K}$$

and by taking union bound, the result holds with probability at least $1 - \delta$. Recall the setting of $\beta \sim \widetilde{\mathcal{O}}(H\sqrt{d})$ as in Theorem 4.2, let $K = \widetilde{\mathcal{O}}(H^5 d^2 \epsilon^{-2})$, the policy error $\mathbb{E}_{s \sim \mu}[V_1^*(s; r) - V_1^\pi(s; r)]$ is bounded by $\epsilon$. □

## A.2 Proof of Corollary 4.4

*Proof of Corollary 4.4.* Following the proof of Theorem 4.2, since for all $\mathbf{x} \in \mathbb{R}^d, \|\mathbf{x}\|_1 \leq \|\mathbf{x}\|_2 \leq \sqrt{d}\|\mathbf{x}\|_1$ it follows that

$$\|\boldsymbol{\psi}_{V_{h+1}}(s_h, \pi_h(s_h))\|_{\boldsymbol{\Sigma}_{1,K+1}^{-1}} = \|\boldsymbol{\Sigma}_{1,K+1}^{-1/2}\boldsymbol{\psi}_{V_{h+1}}(s_h, \pi_h(s_h))\|_2$$
$$\leq \sqrt{d}\|\boldsymbol{\Sigma}_{1,K+1}^{-1/2}\boldsymbol{\psi}_{V_{h+1}}(s_h, \pi_h(s_h))\|_1. \tag{A.7}$$

We denote $\widetilde{u}_h^k$ as the result using the $\ell_1$ norm as the surrogate objective function in this optimization problem (4.5), i.e.

$$\widetilde{u}_h^k := \underset{f \in \mathcal{S} \mapsto [0, H-h]}{\operatorname{argmax}} \|\boldsymbol{\Sigma}_{1,k}^{-1/2} \boldsymbol{\psi}_f(s_h^k, a_h^k)\|_1,$$

then (A.7) yields

$$\begin{aligned}
\|\boldsymbol{\psi}_{V_{h+1}}(s_h, \pi_h(s_h))\|_{\boldsymbol{\Sigma}_{1,K+1}^{-1}} &\leq \sqrt{d} \|\boldsymbol{\Sigma}_{1,K+1}^{-1/2} \boldsymbol{\psi}_{V_{h+1}}(s_h, \pi_h(s_h))\|_1 \\
&\leq \sqrt{d} \|\boldsymbol{\Sigma}_{1,K+1}^{-1/2} \boldsymbol{\psi}_{\widetilde{u}_h^k}(s_h, \pi_h(s_h))\|_1 \\
&\leq \sqrt{d} \|\boldsymbol{\Sigma}_{1,K+1}^{-1/2} \boldsymbol{\psi}_{\widetilde{u}_h^k}(s_h, \pi_h(s_h))\|_2 \\
&\leq \sqrt{d} \|\boldsymbol{\Sigma}_{1,K+1}^{-1/2} \boldsymbol{\psi}_{u_h^k}(s_h, \pi_h(s_h))\|_2,
\end{aligned}$$

where the second inequality comes from $\widetilde{u}_h^k$ is the solution in (4.5), the third inequality comes from the fact that $\|\mathbf{x}\|_1 \leq \|\mathbf{x}\|_2$ and the forth inequality comes from the definition that $u_h^k$. Then (A.3) is changed to be

$$I_1 \leq H\sqrt{d} V_1^\pi(s, \{r_h^k\}_{h=1}^k).$$

Noticing that comparing to the original result, there's an additional $\sqrt{d}$ factor which yields (A.7)

$$\mathbb{E}_{s \sim \mu}[V_1^*(s; r) - V_1^\pi(s; r)] \leq \frac{H\sqrt{d}}{K} \sum_{k=1}^K V_1^k(s_1^k) + H^2 \sqrt{2d \log(3/\delta)/K}.$$

Then it is easy to show that using $\ell_1$ as the surrogate objective function, the sample complexity of Algorithm 2 turns out to be $\widetilde{\mathcal{O}}(H^5 d^3 \epsilon^{-2})$ □

## A.3 Proof of Theorem 5.1

We are going to analyze Algorithm 3 and provide the proof of Theorem 5.1. Following the proof of Theorem 4.2, we only need to revise Lemmas A.2 and A.3 to continue the proof of Theorem 5.1.

**Lemma A.4** (Confidence interval, Bernstein). Let $\beta, \widehat{\beta}, \widetilde{\beta}, \check{\beta}$ and $\lambda$ be defined as Theorem 5.1, then with probability at least $1 - \delta/3$, for all $k \in [K+1]$,

$$\|\boldsymbol{\theta}^* - \widehat{\boldsymbol{\theta}}_k\|_{\widehat{\boldsymbol{\Sigma}}_{1,k}} \leq \widehat{\beta}, \ \|\boldsymbol{\theta}^* - \widehat{\boldsymbol{\theta}}_k\|_{\widehat{\boldsymbol{\Sigma}}_{1,k}} \leq \check{\beta}, \ \|\boldsymbol{\theta}^* - \widetilde{\boldsymbol{\theta}}_k\|_{\widetilde{\boldsymbol{\Sigma}}_{1,k}} \leq \widetilde{\beta}, \ \|\boldsymbol{\theta}^* - \boldsymbol{\theta}_{K+1}\|_{\boldsymbol{\Sigma}_{1,K+1}} \leq \beta,$$

$$(A.8)$$

and $|[\mathbb{V}_h V_{h+1}^k](s, a) - \bar{\mathbb{V}}_h^k(s, a)| \leq E_h^k(s, a)$.

**Lemma A.5** (Summation, Bernstein). For Algorithm 2, setting its parameters as in Lemma A.2, with probability at least $1 - \delta/3$, the summation of the value function during exploration phase is controlled by

$$\sum_{k=1}^K V_1^k(s_1^k) \leq \widetilde{\mathcal{O}}(\sqrt{H^3 K d} + H d\sqrt{K}) + o(\sqrt{K}).$$

*Proof of Theorem 5.1.* The proof is almost the same as the proof of Theorem 4.2 by replacing Lemma A.2 with Lemma A.4, Lemma A.3 with Lemma A.5. In detail, following the same method, (A.6) works for Algorithm 3 under the condition in Lemma A.4 holds. Therefore, by using Lemma A.5 instead of Lemma A.3, with probability at least $1 - \delta$,

$$\begin{aligned}
\mathbb{E}_{s \sim \mu}[V_1^*(s; r) - V_1^\pi(s; r)] &\leq \frac{H}{K} \sum_{k=1}^K V_1^k(s_1^k) + H^2 \sqrt{2 \log(3/\delta)/K} \\
&\leq \widetilde{\mathcal{O}}\Big((\sqrt{H^4 d^2} + \sqrt{H^5 d})/\sqrt{K}\Big).
\end{aligned}$$

Letting $K = \widetilde{\mathcal{O}}(H^4 d(H+d)\epsilon^{-2})$, the policy error for the planning phase could be controlled by $\mathbb{E}_{s \sim \mu}[V_1^*(s; r) - V_1^\pi(s; r)] \leq \epsilon$. □

## A.4 Proof of Corollary 5.3

*Proof of Corollary 5.3.* The proof is almost the same as proof of Corollary 4.4, by adding the additional dependency $d$ into the regret bound achieved by Theorem 5.1, it's easy to verify that the sample complexity using the $\ell_1$ norm as the surrogate function (4.5) is $\widetilde{\mathcal{O}}(H^4 d^2(H+d)\epsilon^{-2})$ $\qquad\square$

# B  Missing Proofs in Appendix A

## B.1  Filtration

For the simplicity of further proof, we define the event filtration here as

$$\mathcal{G}_{h,k} = \big\{\{s_i^\kappa, a_i^\kappa\}_{i=1,\kappa=1}^{H,k-1}, \{s_i^k, a_i^k\}_{i=1}^{h-1}\big\},$$

it is easy to verify that $s_h^k$ is $\mathcal{G}_{h+1,k}$-measurable. Also, since $\pi^k$ is $\mathcal{G}_{h,k}$-measurable for all $h \in [H]$, $a_h^k = \pi_h^k(s_h^k)$ is also $\mathcal{G}_{h+1,k}$-measurable. Also, for any function $f \le R$ built on $\mathcal{G}_{h+1,k}$, such as $V_{h+1}^k, u_h^k, f(s_{h+1}^k) - [\mathbb{P}f](s_h^k, a_h^k)$ is $\mathcal{G}_{h+1,k}$-measurable and it is also a zero-mean $R$-sub-Gaussian conditioned on $\mathcal{G}_{h+1,k}$.

Since $\mathcal{G}_{H+1,k} = \mathcal{G}_{1,k+1}$, we could arrange the filtration as

$$\mathcal{G} = \{\mathcal{G}_{1,1}, \cdots, \mathcal{G}_{H,1}, \cdots, \mathcal{G}_{1,k}, \cdots, \mathcal{G}_{h,k}, \cdots \mathcal{G}_{H,k}, \cdots, \mathcal{G}_{1,k+1}, \cdots, \mathcal{G}_{H,K}, \mathcal{G}_{1,K+1}\},$$

and we will use $\mathcal{G}$ as the filtration set for all of the proofs in the following section and it is obvious that $\mathcal{G}_{1,K+1}$ contains all information we collect during the exploration phase.

## B.2  Proof of Lemma A.1

*Proof of Lemma A.1.* We prove this lemma by induction on time step $h$. Indeed, when $h = H + 1$, $V_{H+1}(s) = V_{H+1}^*(s;r) = 0$ by definition. Suppose for $h \in [H]$, $V_{h+1}(s) \ge V_{h+1}^*(s;r)$, then following the update rule of $Q$ function in Algorithm 1, we have

$$Q_h(s,a) - Q_h^*(s,a;r)$$
$$= \min\big\{H, r_h(s,a) + \langle \boldsymbol{\psi}_{V_{h+1}}(s,a), \boldsymbol{\theta}\rangle + \beta\|\boldsymbol{\psi}_{V_{h+1}}(s,a)\|_{\boldsymbol{\Sigma}^{-1}}\big\} - r_h(s,a) - [\mathbb{P}V_{h+1}^*](s,a;r)$$
$$\ge \min\big\{H - Q_h^*(s,a;r), \langle \boldsymbol{\psi}_{V_{h+1}}(s,a), \boldsymbol{\theta}\rangle + \beta\|\boldsymbol{\psi}_{V_{h+1}}(s,a)\|_{\boldsymbol{\Sigma}^{-1}} - [\mathbb{P}V_{h+1}^*](s,a;r)\big\}.$$

We need to show that $Q_h(s,a) \ge Q_h^*(s,a;r)$. Since it is obvious that the first term $H - Q_h^*(s,a;r)$ in min operator is greater than zero, we only need to verify that the second term is also positive where

$$\langle \boldsymbol{\psi}_{V_{h+1}}(s,a), \boldsymbol{\theta}\rangle + \beta\|\boldsymbol{\psi}_{V_{h+1}}(s,a)\|_{\boldsymbol{\Sigma}^{-1}} - [\mathbb{P}V_{h+1}^*](s,a;r)$$
$$\ge \langle \boldsymbol{\psi}_{V_{h+1}}(s,a), \boldsymbol{\theta}\rangle + \beta\|\boldsymbol{\psi}_{V_{h+1}}(s,a)\|_{\boldsymbol{\Sigma}^{-1}} - [\mathbb{P}V_{h+1}](s,a;r)$$
$$= \langle \boldsymbol{\psi}_{V_{h+1}}(s,a), \boldsymbol{\theta} - \boldsymbol{\theta}^*\rangle + \beta\|\boldsymbol{\psi}_{V_{h+1}}(s,a)\|_{\boldsymbol{\Sigma}^{-1}}$$
$$\ge \beta\|\boldsymbol{\psi}_{V_{h+1}}(s,a)\|_{\boldsymbol{\Sigma}^{-1}} - \|\boldsymbol{\psi}_{V_{h+1}}(s,a)\|_{\boldsymbol{\Sigma}^{-1}}\|\boldsymbol{\theta} - \boldsymbol{\theta}^*\|_{\boldsymbol{\Sigma}},$$

where the first inequality is from the induction assumption that $V_{h+1}^*(s;r) \le V_{h+1}(s)$. The second equality is from the expectation of value function is a linear function of $\boldsymbol{\psi}_{V_{h+1}}$ shown in (3.2). Then the inequality on the third line is utilizing the fact that $\langle \mathbf{x}, \mathbf{y}\rangle \ge -\|\mathbf{x}\|_{\mathbf{A}^{-1}}\|\mathbf{y}\|_{\mathbf{A}}$. Since it is guaranteed that $\beta \ge \|\boldsymbol{\theta} - \boldsymbol{\theta}^*\|_{\boldsymbol{\Sigma}}$ from the statement of this lemma, $Q_h(s,a) - Q_h^*(s,a;r) \ge 0$, which from induction we get our conclusion.

For the second part controlling $V_1(s) - V_1^\pi(s)$, since aforementioned proof has shown that $V_h^*(s;r) \le V_h(s)$ for all $h \in [H]$, we have $V_h^*(s;r) - V_h^\pi(s;r) \le V_h(s) - V_h^\pi(s;r)$ and

$$V_h(s) - V_h^\pi(s;r) = \min\{H, r_h(s, \pi_h(s)) + \langle \boldsymbol{\psi}_{V_{h+1}}, \boldsymbol{\theta}\rangle + \beta\|\boldsymbol{\psi}_{V_{h+1}}(s, \pi_h(s))\|_{\boldsymbol{\Sigma}^{-1}}\}$$
$$- r_h(s, \pi_h(s)) - [\mathbb{P}V_{h+1}^\pi](s, \pi_h(s); r)$$
$$\le \min\{H, \langle \boldsymbol{\psi}_{V_{h+1}}, \boldsymbol{\theta}\rangle + \beta\|\boldsymbol{\psi}_{V_{h+1}}(s, \pi_h(s))\|_{\boldsymbol{\Sigma}^{-1}} - [\mathbb{P}V_{h+1}](s, \pi_h(s))\}$$
$$+ [\mathbb{P}V_{h+1}](s, \pi_h(s))\} - [\mathbb{P}V_{h+1}^\pi](s, \pi_h; r)$$
$$= \min\{H, \langle \boldsymbol{\psi}_{V_{h+1}}, \boldsymbol{\theta} - \boldsymbol{\theta}^*\rangle + \beta\|\boldsymbol{\psi}_{V_{h+1}}(s, \pi_h(s))\|_{\boldsymbol{\Sigma}^{-1}}\}$$
$$+ [\mathbb{P}V_{h+1}](s, \pi_h(s))\} - [\mathbb{P}V_{h+1}^\pi](s, \pi_h(s); r)$$
$$\le \min\{H, 2\beta\|\boldsymbol{\psi}_{V_{h+1}}(s, \pi_h(s))\|_{\boldsymbol{\Sigma}^{-1}}\}$$
$$+ [\mathbb{P}V_{h+1}](s, \pi_h(s))\} - [\mathbb{P}V_{h+1}^\pi](s, \pi_h(s); r),$$

where the first inequality is directly from moving term $-r_h(s, \pi_h(s)) - [\mathbb{P}V_{h+1}](s, \pi_h(s))$ into the $\min$ operator, the second inequality uses the condition that $\|\boldsymbol{\theta} - \boldsymbol{\theta}^*\|_{\boldsymbol{\Sigma}} \leq \beta$ and $\langle \mathbf{x}, \mathbf{y} \rangle \leq \|\mathbf{x}\|_{\mathbf{A}^{-1}} \|\mathbf{y}\|_{\mathbf{A}}$. Considering the first step $h = 1$, we have

$$
\begin{aligned}
V_1(s_1) - V_1^\pi(s_1; r) \leq{} & \min\{H, 2\beta\|\boldsymbol{\psi}_{V_2}(s_1, \pi_1(s_1))\|_{\boldsymbol{\Sigma}^{-1}}\} + \mathbb{E}_{s_2 \sim \mathbb{P}(\cdot|s_1, \pi_1(s_1))}[V_2(s_2) - V_2^\pi(s_2)] \\
\leq{} & \min\{H, 2\beta\|\boldsymbol{\psi}_{V_2}(s_1, \pi_1(s_1))\|_{\boldsymbol{\Sigma}^{-1}}\} \\
& + \mathbb{E}_{s_2 \sim \mathbb{P}(\cdot|s_1, \pi_1(s_1))}\Big[ \min\{H, 2\beta\|\boldsymbol{\psi}_{V_3}(s_2, \pi_2(s_2))\|_{\boldsymbol{\Sigma}^{-1}}\} \\
& + \mathbb{E}_{s_3 \sim \mathbb{P}(\cdot|s_2, \pi_2(s_2))}[V_3(s_3) - V_3^\pi(s_3)]\Big] \\
\leq{} & \cdots \\
\leq{} & \mathbb{E}\bigg[ \sum_{h=1}^H \min\{H, 2\beta\|\boldsymbol{\psi}_{V_{h+1}}(s_h, \pi_h(s_h))\|_{\boldsymbol{\Sigma}^{-1}}\} \bigg| s_1, \pi \bigg],
\end{aligned}
$$

which concludes our proof. $\qquad\square$

### B.3   Proof of Lemma A.2

We introduce the classical confidence set lemma from [1].

**Lemma B.1** (Theorem 2, [1]). *Let $\{\mathcal{F}_t\}_{t=0}^\infty$ be a filtration and $\{\eta_t\}$ is a real-valued stochastic process which is $F_t$-measurable and conditionally $R$-sub-Gaussian. Set $y_t = \langle \mathbf{x}_t, \boldsymbol{\psi}^* \rangle + \eta_t$, $\mathbf{V}_t = \lambda\mathbf{I} + \sum_{i=1}^t \mathbf{x}_i\mathbf{x}_i^\top$ where $\mathbf{x} \in \mathbb{R}^d$. Denote the estimation of $\boldsymbol{\psi}^*$ as $\boldsymbol{\psi}_t = \mathbf{V}_t^{-1}\sum_{i=1}^t y_i\mathbf{x}_i$. If $\|\boldsymbol{\psi}^*\|_2 \leq S, \|\mathbf{x}_t\|_2 \leq L$, then with probability at least $1 - \delta$, for all $t \geq 0$*

$$
\|\boldsymbol{\psi}^* - \boldsymbol{\psi}_t\|_{\mathbf{V}_t} \leq R\sqrt{d\log\left(\frac{1 + tL^2/\lambda}{\delta}\right)} + S\sqrt{\lambda}.
$$

Equipped with this lemma, we begin our proof.

*Proof of Lemma A.2.* Since $[\mathbb{P}u_h^k](s_h^k, a_h^k) = \langle \boldsymbol{\psi}_{u_h^k}(s_h^k, a_h^k), \boldsymbol{\theta}^* \rangle$ due to (3.2) and $u_h^k(s) \leq H$, $u_h^k(s) - \langle \boldsymbol{\psi}_{u_h^k}(s_h^k, a_h^k), \boldsymbol{\theta}^* \rangle$ is $\mathcal{G}_{h,k}$-measurable and it is also a zero mean $H$-sub-Gaussian random variable conditioned on $\mathcal{G}_{h,k}$. Also from Definition 3.1, $\|\boldsymbol{\theta}^*\|_2 \leq B, \|\boldsymbol{\psi}_{u_h^k}(s_h^k, a_h^k)\|_2 \leq H$. Therefore, recall the calculation of $\boldsymbol{\theta}_k$, according to Lemma B.1, let $t = (k-1)H$ we have

$$
\|\boldsymbol{\theta}_k - \boldsymbol{\theta}^*\|_{\boldsymbol{\Sigma}_{1,k}} \leq H\sqrt{d\log\left(\frac{1 + (k-1)H^3/\lambda}{\delta}\right)} + B\sqrt{\lambda}.
$$

Let $\lambda = B^{-2}$, $\delta = \delta/3$ and relax $k$ with $k = K + 1$, we can get the $\beta$ claimed in Theorem 4.2.  $\square$

### B.4   Proof of Lemma A.3

We provide the proof to control the summation of the value function during the exploration phase. To start with, since rather than immediately updating the parameter after each time step, we can only update the estimation $\boldsymbol{\theta}$ and its 'covariance matrix' $\boldsymbol{\Sigma}$ once after each episode. As a result, this 'batched update rule' make the UCB bonus term at step $(h, k)$ be $\|\boldsymbol{\psi}_{u_h^k}(s_h^k, a_h^k)\|_{\mathbf{U}_{1,k}^{-1}}$ instead of $\|\boldsymbol{\psi}_{u_h^k}(s_h^k, a_h^k)\|_{\mathbf{U}_{h,k}^{-1}}$ in the vanilla linear bandit setting. Therefore, we need lemmas showing that these two UCB terms are close to each other.

**Lemma B.2.** *For any $\{\mathbf{x}_{h,k}\}_{h=1,k=1}^{H,K} \subset \mathbb{R}^d$ satisfying that $\|\mathbf{x}_{h,k}\|_2 \leq L, \forall(h,k) \in [H] \times [K]$, let $\mathbf{U}_{h,k} = \lambda\mathbf{I} + \sum_{\kappa=1}^{k-1}\sum_{i=1}^H \mathbf{x}_{i,\kappa}\mathbf{x}_{i,\kappa}^\top + \sum_{i=1}^{h-1}\mathbf{x}_{i,k}\mathbf{x}_{i,k}^\top$, there exists at most $2Hd\log(1 + KHL^2/\lambda)$ pairs of $(h, k)$ tuple such that $\det \mathbf{U}_{h,k} \leq 2\det \mathbf{U}_{1,k}$.*

**Lemma B.3** (Lemma 12, [1]). *Suppose $\mathbf{A}, \mathbf{B} \in \mathbb{R}^{d \times d}$ are two positive definite matrices satisfying that $\mathbf{A} \succeq \mathbf{B}$, then for any $\mathbf{x} \in \mathbb{R}^d$, we have $\|\mathbf{x}\|_{\mathbf{A}} \leq \|\mathbf{x}\|_{\mathbf{B}}\sqrt{\det(\mathbf{A})/\det(\mathbf{B})}$.*

Following that, we also need to introduce the classical lemma to control the summation of the UCB bonus terms in vanilla linear bandit setting.

**Lemma B.4** (Lemma 11, [1]). For any $\{\mathbf{x}_t\}_{t=1}^T \subset \mathbb{R}^d$ satisfying that $\|\mathbf{x}_t\|_2 \leq L, \forall t \in [T]$, let $\mathbf{U}_t = \lambda \mathbf{I} + \sum_{\tau=1}^{t-1} \mathbf{x}_\tau \mathbf{x}_\tau^\top$, we have

$$\sum_{t=1}^T \min\{1, \|\mathbf{x}_t\|_{\mathbf{U}_t^{-1}}\}^2 \leq 2d \log\left(\frac{d\lambda + TL^2}{d\lambda}\right).$$

We also need to introduce the Azuma-Hoeffding's inequality to build the concentration bound for martingale difference sequences.

**Lemma B.5** (Azuma-Hoeffding's inequality, [4]). Let $\{x_i\}_{i=1}^n$ be a martingale difference sequence with respect to a filtration $\{\mathcal{G}_i\}_{i=1}^n$ (i.e. $\mathbb{E}[x_i|\mathcal{G}_i] = 0$ a.s. and $x_i$ is $\mathcal{G}_{i+1}$ measurable) such that $|x_i| \leq M$ a.s.. Then for any $0 < \delta < 1$, with probability at least $1 - \delta$, $\sum_{i=1}^n x_i \leq M\sqrt{2n\log(1/\delta)}$.

*Proof of Lemma A.3.* By Lemma A.1, for the $k$-th episode, we have

$$V_1^k(s_1^k) - V^{\pi^k}(s_1^k) = \mathbb{E}\left[\sum_{h=1}^H \min\{H, 2\beta\|\boldsymbol{\psi}_{V_{h+1}^k}(s_h, \pi_h^k(s_h))\|_{\boldsymbol{\Sigma}_{1,k}^{-1}}\}\Bigg| s_1^k, \pi^k\right]$$

$$\leq \mathbb{E}\left[\sum_{h=1}^H \min\{H, 2\beta\|\boldsymbol{\psi}_{u_h^k}(s_h, \pi_h^k(s_h))\|_{\boldsymbol{\Sigma}_{1,k}^{-1}}\}\Bigg| s_1^k, \pi^k\right] \qquad (B.1)$$

where the inequality comes from that the pseudo value function $u_h^k$ defined in (4.3) is from maximizing the UCB term $\|\boldsymbol{\psi}_{V_{h+1}^k}(s_h, \pi_h^k(s_h))\|_{\boldsymbol{\Sigma}_{1,k}^{-1}}$ and we denote $\{\pi_h^k\}_{h=1}^H$ by $\pi^k$ in short. By the definition of $r_h^k$, we have

$$V^{\pi^k}(s_1^k) = \mathbb{E}[\sum_{h=1}^H r_h^k(s_h, \pi_h^k(s_h))|s_1^k, \pi^k]$$

$$= \mathbb{E}\left[\sum_{h=1}^H \min\{1, 2\beta\|\boldsymbol{\psi}_{u_h^k}(s_h, \pi_h^k(s_h))\|_{\boldsymbol{\Sigma}_{1,k}^{-1}}/H\}\Bigg| s_1^k, \pi^k\right]. \qquad (B.2)$$

Adding (B.1) and (B.2) together and taking summation over $k$, we have

$$\sum_{k=1}^K V_1^k(s_1^k) \leq \frac{H+1}{H}\underbrace{\sum_{k=1}^K \mathbb{E}\left[\sum_{h=1}^H \min\{H, 2\beta\|\boldsymbol{\psi}_{u_h^k}(s_h, \pi_h^k(s_h))\|_{\boldsymbol{\Sigma}_{1,k}^{-1}}\}\Bigg| s_1^k, \pi^k\right]}_{I_1} \leq 2I_1, \qquad (B.3)$$

where the last inequality is due to $(H+1)/H \leq 2$. Next we are going to control the expectation of summation $I_1$. Consider the filtration $\{\mathcal{G}_{h,k}\}_{h=1,k=1}^{H,K}$ defined in Section B.1, denote $x_{h,k}$ as follows:

$$x_{h,k} = \min\{H, 2\beta\|\boldsymbol{\psi}_{u_h^k}(s_h^k, a_h^k)\|_{\boldsymbol{\Sigma}_{1,k}^{-1}}\} - \mathbb{E}_{s_h}\left[\min\{H, 2\beta\|\boldsymbol{\psi}_{u_h^k}(s_h, \pi_h^k(s_h))\|_{\boldsymbol{\Sigma}_{1,k}^{-1}}\}\right],$$

then $x_{h,k}$ is obviously a martingale difference sequence bounded by $H$ w.r.t. $\{\mathcal{G}_{h,k}\}_{h=1,k=1}^{H,K}$. Thus by Azuma-Hoeffding's inequality in Lemma B.5, we have with probability at least $1 - \delta$, $\sum_{k=1}^K \sum_{h=1}^H x_h \leq H\sqrt{2HK\log(1/\delta)}$. Therefore,

$$I_1 = \sum_{k=1}^K \sum_{h=1}^H \min\{H, 2\beta\|\boldsymbol{\psi}_{u_h^k}(s_h^k, a_h^k)\|_{\boldsymbol{\Sigma}_{1,k}^{-1}}\} + \sum_{k=1}^K \sum_{h=1}^H x_h$$

$$\leq 2\beta \sum_{k=1}^K \sum_{h=1}^H \min\{1, \|\boldsymbol{\psi}_{u_h^k}(s_h^k, a_h^k)\|_{\boldsymbol{\Sigma}_{1,k}^{-1}}\} + H\sqrt{2HK\log(1/\delta)}$$

$$\leq 2\sqrt{2}\beta \underbrace{\sum_{k=1}^K \sum_{h=1}^H \min\{1, \|\boldsymbol{\psi}_{u_h^k}(s_h^k, a_h^k)\|_{\boldsymbol{\Sigma}_{h,k}^{-1}}\}}_{I_2} + 4\beta Hd\log(1 + KH^3/\lambda) + H\sqrt{2HK\log(1/\delta)},$$

where the inequality on the second line is due to $2\beta \geq 2H\sqrt{d\log 3} \geq H$ and the last inequality uses Lemma B.3 with $\mathbf{\Sigma}_{1,k}^{-1} \succeq \mathbf{\Sigma}_{h,k}^{-1}$ and $\det \mathbf{\Sigma}_{1,k}^{-1} \leq 2\det \mathbf{\Sigma}_{1,k}^{-1}$ expect for $\widetilde{\mathcal{O}}(Hd)$ cases by Lemma B.2. By $\min\{1, \|\boldsymbol{\psi}_{u_h^k}(s_h, \pi_h^k(s_h))\|_{\mathbf{\Sigma}_{h,k}^{-1}}\} \leq 1$ and $\|\boldsymbol{\psi}_{u_h^k}(s_h^k, a_h^k)\|_2 \leq H$ since $u_h^k \leq H$, we can further bound the $\widetilde{\mathcal{O}}(Hd)$ terms where $\det \mathbf{\Sigma}_{1,k}^{-1} > 2\det \mathbf{\Sigma}_{1,k}^{-1}$. To bound $I_2$, by Lemma B.4, using Cauchy-Schwarz inequality we have

$$I_2 \leq \sqrt{KH}\sqrt{\sum_{k=1}^{K}\sum_{h=1}^{H}\min\{1, \|\boldsymbol{\psi}_{u_h^k}(s_h^k, a_h^k)\|_{\mathbf{\Sigma}_{h,k}^{-1}}^2\}} \leq \sqrt{2KHd\log(1 + KH^3/(d\lambda))},$$

Plugging $I_2$ into $I_1$ then plugging $I_1$ into (B.3). Let $\lambda = B^{-2}$, the summation of the value function $V_1^k(s_1^k)$ is bounded by

$$\sum_{k=1}^{K} V_1^k(s_1^k) \leq 8\beta\Big(\sqrt{HKd\log(1 + KH^3B^2/d)} + dH\log(1 + KH^3B^2)\Big)$$
$$+ 2H\sqrt{2HK\log(1/\delta)}.$$

Taking $\delta = \delta/3$, we can finalize the proof of Lemma A.3. $\qquad\square$

## B.5 Proof of Lemma A.4

The proof of this lemma is similar to the proof of Lemma 5.2 in [29]. We extend their proof to a *time varying* reward and *homogeneous* setting, where the rewards (i.e., the exploration-driven reward function $r_h^k$) are different in different episode $k$. To prove this lemma, we need to introduce the Bernstein inequality for vector-valued martingales.

**Lemma B.6** (Theorem 4.1, [29]). Let $\{\mathcal{G}_t\}_{t=1}^{\infty}$ be a filtration, $\{\mathbf{x}_t, \eta_t\}_{t\geq 1}$ a stochastic process so that $\mathbf{x}_t \in \mathbb{R}^d$ is $\mathcal{G}_t$-measurable and $\eta_t$ is $\mathcal{G}_{t+1}$-measurable. Fix $R, L, \sigma, \lambda > 0, \boldsymbol{\mu}^* \in \mathbb{R}^d$. For $t \geq 1$, let $y_t = \langle\boldsymbol{\mu}^*, \mathbf{x}_t\rangle + \eta_t$. Suppose $\eta_t, \mathbf{x}_t$ satisfy

$$|\eta_t| \leq R, \ \mathbb{E}[\eta_t|\mathcal{G}_t] = 0, \ \mathbb{E}[\eta_t^2|\mathcal{G}_t] \leq \sigma^2, \ \|\mathbf{x}_t\|_2 \leq L.$$

Then for any $0 < \delta < 1$, with probability at least $1 - \delta$, we have

$$\forall t > 0, \ \Big\|\sum_{\tau=1}^{t}\mathbf{x}_\tau\eta_\tau\Big\|_{\mathbf{U}_\tau^{-1}} \leq \beta_t, \ \|\boldsymbol{\mu}_t - \boldsymbol{\mu}^*\|_{\mathbf{U}_t} \leq \beta_t + \sqrt{\lambda}\|\boldsymbol{\mu}^*\|_2,$$

where $\boldsymbol{\mu}_t = \mathbf{U}_t^{-1}\mathbf{b}_t, \mathbf{U}_t = \lambda\mathbf{I} + \sum_{\tau=1}^{t}\mathbf{x}_\tau\mathbf{x}_\tau^\top, \mathbf{b}_t = \sum_{\tau=i}^{t} y_\tau\mathbf{x}_\tau$, and

$$\beta_t = 8\sigma\sqrt{d\log(1 + tL^2/d\lambda)\log(4t^2/\delta)} + 4R\log(4t^2/\delta).$$

We also introduce the following lemma to analyze the error between the estimated variance $\bar{\mathbb{V}}_h^k$ and the true variance $\mathbb{V}_h^k$.

**Lemma B.7** (Lemma C.1, [29]). Let $\mathbb{V}_h^k(s, a)$ be as defined in (3.1) and $\bar{\mathbb{V}}_h^k(s, a)$ be as defined in (5.3), then

$$|\mathbb{V}_h^k(s, a) - \bar{\mathbb{V}}_h^k(s, a)| \leq \min\Big\{H^2, \|\boldsymbol{\psi}_{[V_{h+1}^k]^2}(s, a)\|_{\widetilde{\mathbf{\Sigma}}_{1,k}^{-1}}\|\widetilde{\boldsymbol{\theta}}_k - \boldsymbol{\theta}^*\|_{\widetilde{\mathbf{\Sigma}}_{1,k}}\Big\}$$
$$+ \min\Big\{H^2, 2H\|\boldsymbol{\psi}_{V_{h+1}^k}(s, a)\|_{\widehat{\mathbf{\Sigma}}_{1,k}^{-1}}\|\widehat{\boldsymbol{\theta}}_k - \boldsymbol{\theta}^*\|_{\widehat{\mathbf{\Sigma}}_{1,k}}\Big\}.$$

Equipped with these lemmas, we can start the proof of Lemma A.4.

*Proof of Lemma A.4.* Recall the regression in (5.4). For the regression on $\widehat{\mathbf{\Sigma}}, \widehat{\boldsymbol{\theta}}$, let $\mathbf{x}_h^k = \boldsymbol{\psi}_{V_{h+1}^k}(s_h^k, a_h^k)/\bar{\sigma}_h^k$, and $\eta_h^k = V_{h+1}^k(s_{h+1}^k)/\bar{\sigma}_h^k - \langle\boldsymbol{\theta}^*, \mathbf{x}_h^k\rangle$. Since $\bar{\sigma}_h^k \geq H/\sqrt{d}$ defined in (5.2), we get $\|\mathbf{x}_h^k\|_2 \leq \sqrt{d}, |\eta_h^k| \leq \sqrt{d}$, thus one could verify that $\mathbb{E}[[\eta_h^k]^2|\mathcal{G}_{h,k}] \leq d, \mathbb{E}[\eta_h^k|\mathcal{G}_{h,k}] = 0$, from Lemma B.6, taking $t = (k-1)H$ we have

$$\|\boldsymbol{\theta}^* - \widehat{\boldsymbol{\theta}}_k\|_{\widehat{\mathbf{\Sigma}}_{1,k}} \leq 8d\sqrt{\log(1 + (k-1)H/\lambda)\log(4(k-1)^2H^2/\delta)}$$
$$+ 4\sqrt{d}\log(4(k-1)^2H^2/\delta) + \sqrt{\lambda}B.$$

For the regression of $\widetilde{\boldsymbol{\Sigma}}, \widetilde{\boldsymbol{\theta}}$, $\mathbf{x}_h^k = \boldsymbol{\psi}_{[V_{h+1}^k]^2}(s_h^k, a_h^k)$ which directly implies $\|\mathbf{x}_h^k\|_2 \le H^2$. Let $\eta_h^k = V_{h+1}^k(s_{h+1}^k)^2 - \langle \boldsymbol{\theta}^*, \mathbf{x}_h^k \rangle$, one can easily verify that $|\eta_h^k| \le H^2$ and $\mathbb{E}[\eta_h^k|\mathcal{G}_{h,k}] = 0, \mathbb{E}[[\eta_h^k]^2|\mathcal{G}_{h,k}] \le H^4$, thus using Lemma B.6 again we have

$$\|\boldsymbol{\theta}^* - \widetilde{\boldsymbol{\theta}}_k\|_{\widetilde{\boldsymbol{\Sigma}}_{1,k}} \le 8H^2\sqrt{d\log(1 + (k-1)H/\lambda)\log(4(k-1)^2 H^2/\delta)}$$
$$+ 4H^2\log(4(k-1)^2 H^2/\delta) + \sqrt{\lambda}B.$$

Since $\lambda = B^{-2}$, if we select $\check{\beta}$ and $\widetilde{\beta}$ as

$$\check{\beta} = 8d\sqrt{\log(1 + KHB^2/)\log(4K^2 H^2/\delta)} + 4\sqrt{d}\log(4(k-1)^2 H^2/\delta) + 1$$
$$\widetilde{\beta} = 8H^2\sqrt{d\log(1 + KHB^2)\log(4K^2 H^2/\delta)} + 4H^2\log(4K^2 H^2/\delta) + 1,$$

then with probability at least $1 - 2\delta$, for all $k \in [K+1]$, $\|\boldsymbol{\theta}^* - \widehat{\boldsymbol{\theta}}_k\|_{\widehat{\boldsymbol{\Sigma}}_{1,k}} \le \check{\beta}$, $\|\boldsymbol{\theta}^* - \widetilde{\boldsymbol{\theta}}_k\|_{\widetilde{\boldsymbol{\Sigma}}_{1,k}} \le \widetilde{\beta}$.

Next we are going to give the choice of $\widehat{\beta}$ to make sure that $\|\boldsymbol{\theta}^* - \widehat{\boldsymbol{\theta}}_k\|_{\widehat{\boldsymbol{\Sigma}}_{1,k}} \le \check{\beta}$ holds with high probability. The following proof is conditioned on that the aforementioned event $\|\boldsymbol{\theta}^* - \widehat{\boldsymbol{\theta}}_k\|_{\widehat{\boldsymbol{\Sigma}}_{1,k}} \le \check{\beta}$, $\|\boldsymbol{\theta}^* - \widetilde{\boldsymbol{\theta}}_k\|_{\widetilde{\boldsymbol{\Sigma}}_{1,k}} \le \widetilde{\beta}$ holds, then from Lemma B.7 we have

$$|\mathbb{V}_h^k(s,a) - \bar{\mathbb{V}}_h^k(s,a)|$$
$$\le \min\left\{H^2, \widetilde{\beta}\|\boldsymbol{\psi}_{[V_{h+1}^k]^2}(s,a)\|_{\widetilde{\boldsymbol{\Sigma}}_{1,k}^{-1}}\right\} + \min\left\{H^2, 2\check{\beta}H\|\boldsymbol{\psi}_{V_{h+1}^k}(s,a)\|_{\widehat{\boldsymbol{\Sigma}}_{1,k}^{-1}}\right\}$$
$$= E_h^k(s,a) \tag{B.4}$$

Again, let $\mathbf{x}_h^k = \boldsymbol{\psi}_{V_{h+1}^k}(s_h^k, a_h^k)/\bar{\sigma}_h^k$ to denote the context vector and $\eta_h^k = V_{h+1}^k(s_{h+1}^k)/\bar{\sigma}_h^k - \langle \boldsymbol{\theta}^*, \mathbf{x}_h^k \rangle$ to denote the noise term, since $\|\boldsymbol{\theta}^* - \widehat{\boldsymbol{\theta}}_k\|_{\widehat{\boldsymbol{\Sigma}}_{1,k}} \le \check{\beta}$, we have

$$\mathbb{E}[[\eta_h^k]^2|\mathcal{G}_{h,k}] = \mathbb{V}_h^k(s_h^k, a_h^k)/\nu_h^k \le (E_h^k(s_h^k, a_h^k) + \bar{\mathbb{V}}_h^k(s_h^k, a_h^k))/\nu_h^k \le 1,$$

where the first inequality is from (B.4), the second inequality holds because the definition of $\nu_h^k$ in (5.2).

Therefore we have verified that the noise term $\eta_h^k$ is a zero-mean random variable conditioned on $\mathcal{G}_{h,k}$ and $\mathbb{E}[[\eta_h^k]^2|\mathcal{G}_{h,k}] \le 1$. In that case, using Lemma B.6 again we could get with probability at least $1 - \delta$,

$$\|\boldsymbol{\theta}^* - \widehat{\boldsymbol{\theta}}_k\|_{\widehat{\boldsymbol{\Sigma}}_{1,k}} \le 8\sqrt{d(1 + (k-1)H/\lambda)\log(4(k-1)^2 H^2/\delta)} \tag{B.5}$$
$$+ 4\sqrt{d}\log(4(k-1)^2 H^2/\delta) + \sqrt{\lambda}B, \tag{B.6}$$

again, since $\lambda = B^{-2}$, if we select $\widehat{\beta}$ as

$$\widehat{\beta} = 8\sqrt{d(1 + KHB^2)\log(4K^2 H^2/\delta)} + 4\sqrt{d}\log(4K^2 H^2/\delta) + 1,$$

then $\|\boldsymbol{\theta}^* - \widehat{\boldsymbol{\theta}}_k\|_{\widehat{\boldsymbol{\Sigma}}_{1,k}} \le \widehat{\beta}$ with probability at least $1 - \delta$ for all $k \in [K+1]$.

Next, for the regression of $\boldsymbol{\theta}_{K+1}, \boldsymbol{\Sigma}_{1,K+1}$, by Lemma A.2, we obtain the same result with the selection of $\beta$ as

$$\beta = H\sqrt{d\log\left(\frac{1 + KH^3/\lambda}{\delta}\right)} + B\sqrt{\lambda},$$

which suggests that with probability at least $1 - \delta$, $\|\boldsymbol{\theta}_{K+1} - \boldsymbol{\theta}^*\|_{\boldsymbol{\Sigma}_{1,K+1}} \le \beta$. Then taking union bound with all aforementioned event $\|\boldsymbol{\theta}^* - \widehat{\boldsymbol{\theta}}_k\|_{\widehat{\boldsymbol{\Sigma}}_{1,k}} \le \check{\beta}$, $\|\boldsymbol{\theta}^* - \widetilde{\boldsymbol{\theta}}_k\|_{\widetilde{\boldsymbol{\Sigma}}_{1,k}} \le \widetilde{\beta}$, $\|\boldsymbol{\theta}^* - \widehat{\boldsymbol{\theta}}_k\|_{\widehat{\boldsymbol{\Sigma}}_{1,k}} \le \widehat{\beta}$, we have all these events mentioned in this proof holds with probability at least $1 - 4\delta$. Replace $\delta$ with $\delta/12$, we obtain our final results.

Next, for the regression of $\boldsymbol{\theta}_{K+1}, \boldsymbol{\Sigma}_{1,K+1}$, by Lemma A.2, we obtain the same result with the selection of $\beta$ as

$$\beta = H\sqrt{d\log\left(\frac{1 + KH^3/\lambda}{\delta}\right)} + B\sqrt{\lambda},$$

which suggests that with probability at least $1 - \delta$, $\|\boldsymbol{\theta}_{K+1} - \boldsymbol{\theta}^*\|_{\boldsymbol{\Sigma}_{1,K+1}} \le \beta$. Again, taking an additional union bound, with probability at least $1 - 4\delta$, all events mentioned in this proof hold. Replace $\delta$ with $\delta/12$, we obtain our final results. $\qquad\square$

## B.6 Proof of Lemma A.5

The proof of this lemma borrows some intuition from the proof of Theorem 5.3 in [29]. Unlike Zhou et al. [29] that deals the fixed reward and time-inhomogeneous setting, we need to extend their proof in order to deal with the time-varying reward and time-homogeneous setting.

The next lemmas shows the relationship between the summation of $\nu_h^k$ and the difference between $V_h^k(s)$ calculated in Algorithm 3 and $V_h^{\pi^k}(s; \{r_h^k\}_{h=1,k=1}^{H,K})$

**Lemma B.8.** Let $V_h^k, \nu_h^k$ be defined in Algorithm 3. Then if the condition in Lemma A.4 holds, the following inequality holds with probability at least $1 - 2\delta$,

$$
\sum_{k=1}^{K} [V_1^k(s_1^k) - V_1^{\pi^k}(s_1^k)] \le 4\sqrt{d}\widehat{\beta}\sqrt{\sum_{k=1}^{K}\sum_{h=1}^{H} \nu_h^k}\sqrt{\log(1 + KHB^2)}
$$
$$
+ 2H^2 d \log(1 + KHB^2 d) + H\sqrt{2KH\log(1/\delta)}
$$
$$
\sum_{k=1}^{K}\sum_{h=1}^{H} [\mathbb{P}(V_{h+1}^k - V_{h+1}^{\pi^k})](s_h^k, a_h^k) \le 4\sqrt{d}H\widehat{\beta}\sqrt{\sum_{k=1}^{K}\sum_{h=1}^{H} \nu_h^k}\sqrt{\log(1 + KHB^2)}
$$
$$
+ 2H^3 d \log(1 + KHB^2 d) + 2H^2\sqrt{2KH\log(1/\delta)},
$$

**Lemma B.9.** Let $V_h^k, \nu_h^k$ be defined in Algorithm 3. Then if the condition in Lemma A.4 holds, with probability at least $1 - \delta$,

$$
\sum_{k=1}^{K}\sum_{h=1}^{H} \nu_h^k \le \frac{H^3 K}{d} + 3H^2 K + 3H^3 \log(1/\delta) + 2H\sum_{k=1}^{K}\sum_{h=1}^{H} [\mathbb{P}(V_{h+1}^k - V_{h+1}^{\pi^k}](s_h^k, a_h^k)
$$
$$
+ 2\widetilde{\beta}\sqrt{KHd\log(1 + KH^5 B^2/d)} + 4\widetilde{\beta}Hd\log(1 + KH^5 B^2/d)
$$
$$
+ 8H^2\check{\beta}\sqrt{KHd\log(1 + KHB^2)} + 8H^3 d\check{\beta}\log(1 + KHdB^2).
$$

Equipped with these two lemmas, we can start to prove Lemma A.5.

*Proof of Lemma A.5.* In this proof, we use $\widetilde{\mathcal{O}}(\cdot)$ to ignore all constant and log terms to simplify the results. Recall the selection of $\beta, \widehat{\beta}, \check{\beta}, \widetilde{\beta}$, we have $\beta = \widetilde{\mathcal{O}}(H\sqrt{d})$, $\widehat{\beta} = \widetilde{\mathcal{O}}(\sqrt{d})$, $\check{\beta} = \widetilde{\mathcal{O}}(d)$, $\widetilde{\beta} = \widetilde{\mathcal{O}}(H^2\sqrt{d})$. Therefore Lemma B.8 could be simplified as

$$
\sum_{k=1}^{K}\sum_{h=1}^{H} [\mathbb{P}(V_{h+1}^k - V_{h+1}^{\pi^k})](s_h^k, a_h^k) \le \widetilde{\mathcal{O}}\left(Hd\sqrt{\sum_{k=1}^{K}\sum_{h=1}^{H} \nu_h^k} + H^3 d + \sqrt{KH^5}\right). \tag{B.7}
$$

Lemma B.9 could also be simplified as

$$
\sum_{k=1}^{K}\sum_{h=1}^{H} \nu_h^k \le \widetilde{\mathcal{O}}\left(\frac{H^3 K}{d} + H^2 K + H\sum_{k=1}^{K}\sum_{h=1}^{H} [\mathbb{P}(V_{h+1}^k - V_{h+1}^{\pi^k})](s_h^k, a_h^k) + \sqrt{KH^5 d^3} + H^3 d^2\right). \tag{B.8}
$$

Let $\sqrt{\sum_{k=1}^{K}\sum_{h=1}^{H} \nu_h^k} = x$, plugging (B.7) into (B.8), we have

$$
x^2 \le \widetilde{\mathcal{O}}(H^3 K d^{-1} + H^2 K + H^2 dx + H^4 d + \sqrt{KH^7} + \sqrt{KH^5 d^3} + H^3 d^2),
$$

Since the quadratic inequality $x^2 \le \widetilde{\mathcal{O}}(bx + c)$ indicates that $x \le \mathcal{O}(b + \sqrt{c})$, setting

$$
b = \widetilde{\mathcal{O}}(H^2 d), c = \widetilde{\mathcal{O}}(H^3 K d^{-1} + H^2 K + H^4 d + \sqrt{KH^7} + \sqrt{KH^5 d^3} + H^3 d^2),
$$

hence

$$\sqrt{\sum_{k=1}^{K}\sum_{h=1}^{H}\nu_h^k} \leq \widetilde{\mathcal{O}}(H^2 d + \sqrt{H^3 K/d} + H\sqrt{K} + H^2\sqrt{d} + d\sqrt{H^3} + (KH^7)^{1/4} + (KH^5 d^3)^{1/4})$$

(B.9)

$$= \widetilde{\mathcal{O}}(\sqrt{H^3 K/d} + H\sqrt{K}) + o(\sqrt{K}). \tag{B.10}$$

Plugging (B.10) back to Lemma B.8, we have

$$\sum_{k=1}^{K}[V_1^k(s_1^k) - V_1^{\pi^k}(s_1^k)] \leq \widetilde{\mathcal{O}}(\sqrt{H^3 K d} + Hd\sqrt{K}) + o(\sqrt{K}). \tag{B.11}$$

Next we are going to show the bound of the summation over $V_1^{\pi^k}(s_1^k)$, note that this value function is bounded by $H$ and from Bellman equality, we have

$$V_h^{\pi^k}(s_1^k) = r_h^k(s_1^k, a_1^k) + [\mathbb{P}V_{h+1}^{\pi^k}](s_h^k, a_h^k),$$

taking summation over $h \in [H], k \in [K]$ then

$$\sum_{k=1}^{K} V_1^{\pi^k}(s_1^k) = \sum_{k=1}^{K}\sum_{h=1}^{H} r_h^k(s_1^k, a_1^k) + \sum_{k=1}^{K}\sum_{h=1}^{H}[\mathbb{P}V_{h+1}^{\pi^k}](s_h^k, a_h^k) - V_{h+1}^{\pi^k}(s_{h+1}^k)$$

$$\leq \sum_{k=1}^{K}\sum_{h=1}^{H} \min\{1, 2\beta\|\boldsymbol{\psi}_{u_h^k}(s_h^k, a_h^k)\|_{\boldsymbol{\Sigma}_{1,k}^{-1}}/H\} + H\sqrt{HK\log(1/\delta)},$$

where the last inequality holds due to Azuma-Hoeffding's inequality i.e. Lemma B.5. For the first term,

$$\sum_{k=1}^{K}\sum_{h=1}^{H} \min\{1, 2\beta\|\boldsymbol{\psi}_{u_h^k}(s_h^k, a_h^k)\|_{\boldsymbol{\Sigma}_{1,k}^{-1}}/H\} \leq \frac{2\beta}{H}\underbrace{\sum_{k=1}^{K}\sum_{h=1}^{H} \min\{1, \|\boldsymbol{\psi}_{u_h^k}(s_h^k, a_h^k)\|_{\boldsymbol{\Sigma}_{1,k}^{-1}}\}}_{I_1},$$

where the inequality is due to $\beta \geq H\sqrt{\log(12)} \geq H/2$. Using Lemma B.2 and Lemma B.3 with $\boldsymbol{\Sigma}_{1,k}^{-1} \succeq \boldsymbol{\Sigma}_{h,k}^{-1}$ and $\det \boldsymbol{\Sigma}_{1,k}^{-1} \leq 2\det \boldsymbol{\Sigma}_{h,k}^{-1}$ except for $\widetilde{\mathcal{O}}(Hd)$ steps mentioned in Lemma B.2, setting $\lambda = B^{-2}$, we have

$$I_1 \leq 2Hd\log(1 + KH^3 B^2) + \sqrt{2}\sum_{k=1}^{K}\sum_{h=1}^{H} \min\{1, \|\boldsymbol{\psi}_{u_h^k}(s_h^k, a_h^k)\|_{\boldsymbol{\Sigma}_{h,k}^{-1}}\}$$

$$\leq 2Hd\log(1 + KH^3 B^2) + \sqrt{2HK}\sqrt{\sum_{k=1}^{K}\sum_{h=1}^{H} \min\{1, \|\boldsymbol{\psi}_{u_h^k}(s_h^k, a_h^k)\|_{\boldsymbol{\Sigma}_{h,k}^{-1}}^2\}}$$

$$\leq 2Hd\log(1 + KH^3 B^2) + 2\sqrt{HKd\log(1 + KH^3 B^2/d)}.$$

Therefore, since $\beta = \widetilde{\mathcal{O}}(H\sqrt{d})$, then

$$\sum_{k=1}^{K} V_1^{\pi^k}(s_1^k) \leq 4\beta d\log(1 + KH^3 B^2) + 4\beta\sqrt{Kd\log(1 + KH^3 B^2/d)/H} + \sqrt{H^3 K\log(1/\delta)}$$

(B.12)

$$\leq \widetilde{\mathcal{O}}(d\sqrt{KH} + \sqrt{KH^3}) + o(\sqrt{K}). \tag{B.13}$$

Adding (B.11) and (B.13) together, we have the following result,

$$\sum_{k=1}^{K} V_1^k(s_1^k) \leq \widetilde{\mathcal{O}}(\sqrt{H^3 K d} + Hd\sqrt{K}) + o(\sqrt{K}).$$

By taking the union bound, this inequality holds with probability at least $1 - 4\delta$. Since $\delta$ only appears in the logarithmic terms, thus changing $\delta$ to $\delta/12$ will not affect the result. $\square$

# C    Proof of Auxiliary Lemmas in Appendix B

## C.1    Proof of Lemma B.2

*Proof of Lemma B.2.* We want to know how many pairs of $(h,k)$ exists such that $\det \mathbf{U}_{h,k} \geq 2 \det \mathbf{U}_{1,k}$.

Furthermore, we have if there exists $k \in [K]$ such that $\det \mathbf{U}_{1,k+1} \leq 2 \det \mathbf{U}_{1,k}$, then it is obvious that for all $h \in [H]$, we have $\det \mathbf{U}_{h,k} \leq \det \mathbf{U}_{1,k+1} \leq 2 \det \mathbf{U}_{1,k}$.

Therefore, suppose there exists a set $\mathcal{K} \subset [K]$ such that for all $k \notin \mathcal{K}$, $\det \mathbf{U}_{1,k+1} \leq 2 \det \mathbf{U}_{1,k}$ and for all $k \in \mathcal{K}$, $\det \mathbf{U}_{1,k+1} > 2 \det \mathbf{U}_{1,k}$, then the pair of $(h,k)$ such that $\det \mathbf{U}_{h,k} \geq 2 \det \mathbf{U}_{1,k}$ is upper bounded by $H|\mathcal{K}|$.

Notice that for all $k \in \mathcal{K}$, $\det \mathbf{U}_{1,k+1} > 2 \det \mathbf{U}_{1,k}$, it is easy to show that

$$\det \mathbf{U}_{1,K+1} > 2^{|\mathcal{K}|} \det \mathbf{U}_{1,1} = 2^{|\mathcal{K}|} \lambda^d,$$

where the last inequality comes from $\mathbf{U}_{1,1} = \lambda \mathbf{I} \in \mathbb{R}^{d \times d}$. Notice that $\det \mathbf{U} \leq \|\mathbf{U}\|_2^d$, taking log we have

$$d \log(\|\mathbf{U}_{1,K+1}\|_2) \geq \log \det \mathbf{U}_{1,K+1} > |\mathcal{K}| \log 2 + d \log \lambda. \tag{C.1}$$

From the definition of $\mathbf{U}_{1,K+1}$, by triangle inequality,

$$\|\mathbf{U}_{1,K+1}\|_2 \leq \lambda + \sum_{k=1}^{H} K \sum_{h=1}^{H} \|\mathbf{x}_h^k \mathbf{x}_h^{k\top}\|_2 \leq \lambda + KH\|\mathbf{x}_h^k\|_2^2 \leq \lambda + KHL^2, \tag{C.2}$$

where the last inequality is due to $\|\mathbf{x}\|_2 \leq L$ from the statement of the lemma. Therefore we conclude our proof by merging (C.1) and (C.2) together to get

$$|\mathcal{K}| \log 2 < d \log(1 + HKL^2/\lambda),$$

noticing $\log 2 \geq 1/2$ we can get the result claimed in the lemma.                     $\square$

## C.2    Proof of Lemma B.8

*Proof of Lemma B.8.* Assume that the condition in Lemma A.4 holds, then

$$V_h^k(s_h^k) - V_h^{\pi^k}(s_h^k)$$
$$\leq \langle \widehat{\boldsymbol{\theta}}_k, \boldsymbol{\psi}_{V_{h+1}^k}(s_h^k, a_h^k) \rangle - [\mathbb{P}V_{h+1}^{\pi^k}](s_h^k, a_h^k) + \widehat{\beta}\|\boldsymbol{\psi}_{V_{h+1}^k}(s_h^k, a_h^k)\|_{\widehat{\boldsymbol{\Sigma}}_{1,k}^{-1}}$$
$$\leq \|\widehat{\boldsymbol{\theta}}_k - \boldsymbol{\theta}^*\|_{\widehat{\boldsymbol{\Sigma}}_{1,k}} \|\boldsymbol{\psi}_{V_{h+1}^k}(s_h^k, a_h^k)\|_{\widehat{\boldsymbol{\Sigma}}_{1,k}^{-1}} + [\mathbb{P}V_{h+1}^k - V_{h+1}^{\pi^k}](s_h^k, a_h^k) + \widehat{\beta}\|\boldsymbol{\psi}_{V_{h+1}^k}(s_h^k, a_h^k)\|_{\widehat{\boldsymbol{\Sigma}}_{1,k}^{-1}}$$
$$\leq 2\widehat{\beta}\|\boldsymbol{\psi}_{V_{h+1}^k}(s_h^k, a_h^k)\|_{\widehat{\boldsymbol{\Sigma}}_{1,k}^{-1}} + [\mathbb{P}V_{h+1}^k - V_{h+1}^{\pi^k}](s_h^k, a_h^k),$$

where the first inequality holds due to the definition of $V_h^k$, the second inequality holds due to Cauchy-Schwarz inequality and the third one holds due to the condition (A.8) in Lemma A.4. Notice that $V_h^k - V_h^{\pi^k} \leq H$, we have

$$V_h^k(s_h^k) - V_h^{\pi^k}(s_h^k) \leq \min\{H, 2\widehat{\beta}\|\boldsymbol{\psi}_{V_{h+1}^k}(s_h^k, a_h^k)\|_{\widehat{\boldsymbol{\Sigma}}_{1,k}^{-1}}\} + [\mathbb{P}V_{h+1}^k - V_{h+1}^{\pi^k}](s_h^k, a_h^k)$$

Taking summation over $k \in [K]$ and $h \in [H]$, we have

$$\sum_{k=1}^{K}[V_1^k(s_1^k) - V_1^{\pi_k}(s_1^k)] \leq \sum_{k=1}^{K}\sum_{h=1}^{H}\min\{H, 2\widehat{\beta}\|\boldsymbol{\psi}_{V_{h+1}^k}(s_h^k, a_h^k)\|_{\widehat{\boldsymbol{\Sigma}}_{1,k}^{-1}}\}$$
$$+ \sum_{k=1}^{K}\sum_{h=1}^{H}\left[[\mathbb{P}V_{h+1}^k - V_{h+1}^{\pi^k}](s_h^k, a_h^k) - [V_{h+1}^k(s_{h+1}^k) - V_{h+1}^{\pi^k}(s_{h+1}^k)]\right]$$
$$\leq \underbrace{\sum_{k=1}^{K}\sum_{h=1}^{H}\min\{H, 2\widehat{\beta}\|\boldsymbol{\psi}_{V_{h+1}^k}(s_h^k, a_h^k)\|_{\widehat{\boldsymbol{\Sigma}}_{1,k}^{-1}}\}}_{I_1} + H\sqrt{2KH\log(1/\delta)},$$

$$\tag{C.3}$$

where the second inequality is a direct result of Azuma-Hoeffding's inequality as in Lemma B.5.

Next we bound $I_1$. Recall the update rule of $\widehat{\Sigma}_{h,k}$, notice that $\bar{\sigma}_h^k \geq H/\sqrt{d}$ and $\|\psi_{V_{h+1}^k}(s_h^K, a_h^K)\|_2 \leq H$ from $V_{h+1}^k \leq H$, it is easy to verify that $\|\psi_{V_{h+1}^k}(s_h^K, a_h^K)/\widehat{\sigma}_h^k\|_2 \leq \sqrt{d}$. Hence

$$
I_1 \leq \sqrt{2} \sum_{k=1}^{K} \sum_{h=1}^{H} \min\{H, 2\widehat{\beta}\|\psi_{V_{h+1}^k}(s_h^k, a_h^k)\|_{\widehat{\Sigma}_{h,k}^{-1}}\} + 2H^2 d\log(1 + KHd/\lambda)
$$

$$
\leq \sqrt{2}\max\{\sqrt{d}, 2\widehat{\beta}\} \sum_{k=1}^{K} \sum_{h=1}^{H} \bar{\sigma}_h^k \min\{1, \|\psi_{V_{h+1}^k}(s_h^k, a_h^k)/\bar{\sigma}_h^k\|_{\widehat{\Sigma}_{h,k}^{-1}}\} + 2H^2 d\log(1 + KHd/\lambda)
$$

$$
\leq 2\sqrt{2}\widehat{\beta}\sqrt{\sum_{k=1}^{K}\sum_{h=1}^{H}\nu_h^k}\sqrt{\sum_{k=1}^{K}\sum_{h=1}^{H}\min\{1, \|\psi_{V_{h+1}^k}(s_h^k, a_h^k)/\bar{\sigma}_h^k\|_{\widehat{\Sigma}_{h,k}^{-1}}^2\}} + 2H^2 d\log(1 + KHd/\lambda)
$$

$$
\leq 4\widehat{\beta}\sqrt{d}\sqrt{\sum_{k=1}^{K}\sum_{h=1}^{H}\nu_h^k}\sqrt{\log(1 + KH/\lambda)} + 2H^2 d\log(1 + KHd/\lambda),
$$

where the first inequality, similar to the corresponding proof in Lemma A.3, is a direct implication of Lemma B.2 and Lemma B.3 with $\widehat{\Sigma}_{1,k}^{-1} \succeq \widehat{\Sigma}_{h,k}^{-1}$ and $\det \Sigma_{1,k}^{-1} \leq 2\det\widehat{\Sigma}_{1,k}^{-1}$ except for $\widetilde{\mathcal{O}}(Hd)$ cases mentioned in Lemma B.2, the second inequality moves $\bar{\sigma}_h^k$ outside, the third inequality holds because $\widehat{\beta} \geq 4\sqrt{d}\log 12 \geq \sqrt{d}$ and Cauchy-Schwarz inequality, and the forth inequality holds due to Lemma B.4. Plugging $I_1$ into (C.3) and let $h' = 1, \lambda = B^{-2}$, we have

$$
\sum_{k=1}^{K}[V_1^k(s_1^k) - V_1^{\pi^k}(s_1^k)] \leq 4\sqrt{d}\widehat{\beta}\sqrt{\sum_{k=1}^{K}\sum_{h=1}^{H}\nu_h^k}\sqrt{\log(1 + KHB^2)}
$$
$$
+ 2H^2 d\log(1 + KHB^2 d) + H\sqrt{2KH\log(1/\delta)}.
$$

Furthermore, by Azuma-Hoeffding's inequality as in Lemma B.5,

$$
\sum_{k=1}^{K}\sum_{h=1}^{H}\mathbb{P}[V_{h+1}^k - V_{h+1}^{\pi^k}](s_h^k, a_h^k) = \sum_{k=1}^{K}\sum_{h=2}^{H}[V_h^k - V_h^{\pi^k}](s_h^k)
$$
$$
+ \sum_{k=1}^{K}\sum_{h=1}^{H}\left[\mathbb{P}(V_{h+1}^k - V_{h+1}^{\pi^k})(s_h^k, a_h^k) - [V_{h+1}^k - V_{h+1}^{\pi^k}](s_{h+1}^k)\right]
$$
$$
\leq 4\sqrt{d}H\widehat{\beta}\sqrt{\sum_{k=1}^{K}\sum_{h=1}^{H}\nu_h^k}\sqrt{\log(1 + KHB^2)}
$$
$$
+ 2H^3 d\log(1 + KHB^2 d) + (H + 1)H\sqrt{2KH\log(1/\delta)},
$$

which becomes the second part of the statement in the lemma. Using $H + 1 \leq 2H$ we can get the result claimed in the lemma. $\qquad\square$

## C.3   Proof of Lemma B.9

To begin with, we will first show the total variance lemma originally introduced in [8].

**Lemma C.1** (Total variance lemma, Lemma C.5, [8]). [1] With probability at least $1 - \delta$, we have

$$
\sum_{k=1}^{K}\sum_{h=1}^{H}[\mathbb{V}V_h^{\pi^k}(\cdot; \{r_h^k\}_{h=1}^{H})](s, a) \leq 3H^2 K + 3H^3\log(1/\delta).
$$

---

[1] The original Lemma C.5 in Jin et al. [8] holds for the identical reward functions, i.e., $r_h^1 = \cdots = r_h^K$. Their lemma also holds for the general case $r_h^1 \neq \cdots \neq r_h^K$ without changing their proof.

*Proof of Lemma B.9.* Assume the condition in Lemma A.4 holds, we have with probability at least $1 - \delta$,

$$
\sum_{k=1}^{K} \sum_{h=1}^{H} \nu_h^k \le \sum_{k=1}^{K} \sum_{h=1}^{H} \left( \frac{H^2}{d} + \bar{\mathbb{V}}_h^k(s_h^k, a_h^k) + E_h^k(s_h^k, a_h^k) \right)
$$

$$
= \frac{H^3 K}{d} + \underbrace{\sum_{k=1}^{K} \sum_{h=1}^{H} \left( [\mathbb{V}_h V_{h+1}^k](s_h^k, a_h^k) - [\mathbb{V}_h V_{h+1}^{\pi^k}](s_h^k, a_h^k) \right)}_{I_1} + 2 \underbrace{\sum_{k=1}^{H} \sum_{h=1}^{H} E_h^k(s_h^k, a_h^k)}_{I_2}
$$

$$
+ \underbrace{\sum_{k=1}^{K} \sum_{h=1}^{H} [\mathbb{V}_h V_{h+1}^{\pi^k}](s_h^k, a_h^k)}_{I_3} + \underbrace{\sum_{k=1}^{K} \sum_{h=1}^{H} \left[ \bar{\mathbb{V}}_h^k(s_h^k, a_h^k) - [\mathbb{V}_h V_{h+1}^k](s_h^k, a_h^k) - E_h^k \right]}_{I_4}
$$

$$
\le \frac{H^3 K}{d} + I_1 + I_2 + 3H^2 K + 3H^3 \log(1/\delta), \tag{C.4}
$$

where the value function $V_h^{\pi^k}(s)$ is short for $V_h^{\pi^k}(s; \{r_h^k\}_{h=1}^H)$ for simplicity. The first inequality is from the definition of $\nu_h^k$ in (5.2), while the last inequality is from Lemma C.1 to control $I_3$. $I_4 \le 0$ is due to Lemma A.4. Next we are about to bound $I_1$ and $I_2$ separately.

Since the estimated value function $V_{h+1}^k$ and the real value function $V_{h+1}^{\pi^k}$ are both bounded by $[0, H]$, we have

$$
I_1 \le \sum_{k=1}^{K} \sum_{h=1}^{H} \left[ \mathbb{P}([V_{h+1}^k]^2 - [V_{h+1}^{\pi^k}]^2) \right](s_h^k, a_h^k) \le 2H \sum_{k=1}^{K} \sum_{h=1}^{H} [\mathbb{P}(V_{h+1}^k - V_{h+1}^{\pi^k})](s_h^k, a_h^k).
$$

For term $I_2$, we have

$$
I_2 \le \sum_{k=1}^{K} \sum_{h=1}^{H} \min\{H^2, \widetilde{\beta} \| \boldsymbol{\psi}_{[V_{h+1}^k]^2}(s_h^k, a_h^k) \|_{\widetilde{\boldsymbol{\Sigma}}_{1,k}^{-1}} \} + \sum_{k=1}^{K} \sum_{h=1}^{H} \min\{H^2, 2H\check{\beta} \| \boldsymbol{\psi}_{V_{h+1}^k}(s, a) \|_{\widehat{\boldsymbol{\Sigma}}_{1,k}^{-1}} \}
$$

$$
\le \max\{H^2, \widetilde{\beta}\} \sum_{k=1}^{K} \sum_{h=1}^{H} \min\{1, \| \boldsymbol{\psi}_{[V_{h+1}^k]^2}(s_h^k, a_h^k) \|_{\widetilde{\boldsymbol{\Sigma}}_{1,k}^{-1}} \}
$$

$$
+ \sum_{k=1}^{K} \sum_{h=1}^{H} \max\{H^2, 2H\check{\beta}\bar{\sigma}_h^k\} \min\left\{1, \left\| \boldsymbol{\psi}_{V_{h+1}^k}(s, a)/\bar{\sigma}_h^k \right\|_{\widehat{\boldsymbol{\Sigma}}_{1,k}^{-1}} \right\}.
$$

Noticing that from the definition of $\nu_h^k$,

$$
\nu_k^h = \max\{H^2/d, \bar{\mathbb{V}}_h^k(s_h^k, a_h^k) + E_h^k(s_h^k, a_h^k)\} \le \max\{H^2/d, H^2 + 2H^2\} = 3H^2,
$$

thus $\bar{\sigma}_h^k = \sqrt{\nu_h^k} \le 2H$. Recall that $\widetilde{\beta} \ge 4H^2 \log(12) \ge H^2$ and $\check{\beta} \ge 1$, we have

$$
I_2 \le \widetilde{\beta} \underbrace{\sum_{k=1}^{K} \sum_{h=1}^{H} \min\{1, \| \boldsymbol{\psi}_{[V_{h+1}^k]^2}(s_h^k, a_h^k) \|_{\widetilde{\boldsymbol{\Sigma}}_{1,k}^{-1}} \}}_{I_5} + 4H^2 \check{\beta} \underbrace{\sum_{k=1}^{K} \sum_{h=1}^{H} \min\left\{1, \| \boldsymbol{\psi}_{V_{h+1}^k}(s, a)/\bar{\sigma}_h^k \|_{\widehat{\boldsymbol{\Sigma}}_{1,k}^{-1}} \right\}}_{I_6}.
$$

For $I_5$, using Lemmas B.2 and B.3 with $\widetilde{\boldsymbol{\Sigma}}_{1,k}^{-1} \succeq \widetilde{\boldsymbol{\Sigma}}_{h,k}^{-1}$ and $\det \widetilde{\boldsymbol{\Sigma}}_{1,k}^{-1} \le 2 \det \widetilde{\boldsymbol{\Sigma}}_{1,k}^{-1}$ except for $\widetilde{\mathcal{O}}(Hd)$ cases mentioned in Lemma B.2, we have

$$
I_5 \le \sqrt{2} \sum_{k=1}^{K} \sum_{h=1}^{H} \min\{1, \| \boldsymbol{\psi}_{[V_{h+1}^k]^2}(s_h^k, a_h^k) \|_{\widetilde{\boldsymbol{\Sigma}}_{h,k}^{-1}} \} + 2Hd \log(1 + KH^5/d\lambda)
$$

$$
\le \sqrt{2KH} \sqrt{\sum_{k=1}^{K} \sum_{h=1}^{H} \min\{1, \| \boldsymbol{\psi}_{[V_{h+1}^k]^2}(s_h^k, a_h^k) \|_{\widetilde{\boldsymbol{\Sigma}}_{h,k}^{-1}}^2 \}} + 2Hd \log(1 + KH^5/d\lambda)
$$

$$
\le 2\sqrt{KHd \log(1 + KH^5/d\lambda)} + 2Hd \log(1 + KH^5/d\lambda),
$$

where the first inequality is a direct implication from Lemma B.2 and the second inequality is due to Cauchy-Schwarz inequality. The third inequality utilizes Lemma B.4. As for $I_6$, we have

$$
\begin{aligned}
I_6 &\leq \sqrt{2} \sum_{k=1}^{K} \sum_{h=1}^{H} \min\left\{1, \left\|\boldsymbol{\psi}_{V_{h+1}^k}(s,a)/\bar{\sigma}_h^k\right\|_{\widehat{\boldsymbol{\Sigma}}_{h,k}^{-1}}\right\} + 2Hd\log(1 + KHd/\lambda) \\
&\leq \sqrt{2KH} \sqrt{\sum_{k=1}^{K} \sum_{h=1}^{H} \min\left\{1, \left\|\boldsymbol{\psi}_{V_{h+1}^k}(s,a)/\bar{\sigma}_h^k\right\|_{\widehat{\boldsymbol{\Sigma}}_{h,k}^{-1}}\right\} + 2Hd\log(1 + KHd/\lambda)} \\
&\leq 2\sqrt{KHd\log(1 + KH/\lambda)} + 2Hd\log(1 + KHd/\lambda).
\end{aligned}
$$

Finally, plugging $I_5, I_6$ into $I_2$ and $I_1, I_2$ into (C.4) we have

$$
\begin{aligned}
\sum_{k=1}^{K} \sum_{h=1}^{H} \nu_h^k &\leq \frac{H^3 K}{d} + 3H^2 K + 3H^3 \log(1/\delta) + 2H \sum_{k=1}^{K} \sum_{h=1}^{H} [\mathbb{P}(V_{h+1}^k - V_{h+1}^{\pi^k})](s_h^k, a_h^k) \\
&\quad + 2\widetilde{\beta}\sqrt{KHd\log(1 + KH^5/d\lambda)} + 4\widetilde{\beta}Hd\log(1 + KH^5/d\lambda) \\
&\quad + 8H^2\check{\beta}\sqrt{KHd\log(1 + KH/\lambda)} + 8H^3 d\check{\beta}\log(1 + KHd/\lambda).
\end{aligned}
$$

Let $\lambda = B^{-2}$ we could get the result in the statement of the lemma. $\qquad\square$

# D  Proof of Lower Bound

In this section, we will give the detailed proof of the sample complexity lower bound. We start with verifying that the MDP structure as shown in Figure 1 is a linear mixture MDP satisfying Definition 3.1.

## D.1  Verification of the MDP structure

We will first show that the $\ell_2$ norm of $\boldsymbol{\theta}$ is controlled. Recall the $\boldsymbol{\theta}_i$ is set by $\boldsymbol{\Theta} = \left\{\boldsymbol{\theta}_i | \boldsymbol{\theta}_i = \left(\sqrt{2}, \alpha\widetilde{\boldsymbol{\theta}}_i^\top/\sqrt{d}\right)^\top\right\}$ where $\widetilde{\boldsymbol{\theta}}_i = \mathbf{x}_i \in \mathcal{M}$, we can have that $\|\boldsymbol{\theta}_i\|_2 = \sqrt{2 + \alpha^2}$, therefore, as long as the parameter $\alpha$ is an absolute constant, the $\ell_2$ norm of $\boldsymbol{\theta}$ is controlled. Next, considering a function $V \leq D$, we have

$$
\|\boldsymbol{\psi}_V(S_1, a_i)\|_2 = \left\|\left(\frac{V(S_{2,1}) + V(S_{2,2})}{2\sqrt{2}} \quad (V(S_{2,1}) - V(S_{2,2}))\frac{\mathbf{a}_i^\top}{\sqrt{2d}}\right)\right\|_2 \leq \sqrt{\frac{D^2}{2} + \frac{D^2 d}{2d}} = D,
$$

which shows that the MDP structure satisfies Definition 3.1.

## D.2  Proof of Theorem 6.1

We denote the $d-1$-dimension binary vector set as $\mathcal{B}_{d-1} = \{\mathbf{x} | \mathbf{x} \in \mathbb{R}^{d-1}, [\mathbf{x}]_i \in \{-1, 1\}\}$. The next lemma shows that the binary set $\mathcal{M}$ exists.

**Lemma D.1.** Given $\gamma \in (0, 1)$, there exists a $\mathcal{M} \subset \mathcal{B}_{d-1}$ such that for any two different vector $\mathbf{x}, \mathbf{x}' \in \mathcal{M}$, $\langle \mathbf{x}, \mathbf{x}' \rangle \leq (d-1)\gamma$, and the log-cardinality of the proposed set is bounded as $\log(|\mathcal{M}|) < (d-1)\gamma^2/4$.

With this lemma, we can construct a set $\mathcal{M}$ with $|\mathcal{M}| = \lceil \exp(d\gamma^2/4) \rceil - 1$ where $\gamma = \frac{1}{2}$. It is easy to verify that $|\mathcal{M}| < \exp(d\gamma^2/4)$ and

$$
\begin{aligned}
\log(|\mathcal{M}|) &\geq \log(\exp(d\gamma^2/4) - 1) \\
&\geq \frac{(d-1)\gamma^2}{4} + \log(1 - \exp(-(d-1)\gamma^2/4)) \\
&\geq \frac{(d-1)\gamma^2}{4} - 3, \tag{D.1}
\end{aligned}
$$

where the last inequality holds since $\gamma = \frac{1}{2}$ and $d \geq 2$, we have $\log(1 - \exp(-(d-1)\gamma^2/4)) \geq -3$. From Lemma D.1, we know that for any two different vectors $\mathbf{x}, \mathbf{x}' \in \mathcal{M}$, $\langle \mathbf{x}, \mathbf{x}' \rangle \leq (d-1)/2$.

Next lemma establishes the lower bound of sample complexity for any algorithm to estimate the true parameter $\boldsymbol{\theta} \in \Theta$ of the proposed linear mixture MDP, from the sampled state-action pairs of this MDP.

**Lemma D.2.** Suppose an algorithm estimates the underlying parameter $\boldsymbol{\theta}$ by building an estimator $\widehat{\boldsymbol{\theta}}$ from $K$ sampled trajectories. If the algorithm guarantees that $\mathbb{E}_{\boldsymbol{\theta} \sim \mathrm{Unif}\Theta}[\mathbb{1}(\widehat{\boldsymbol{\theta}} = \boldsymbol{\theta})] \geq 1 - \delta$, we have

$$\delta \geq 1 - \left( \frac{d-1}{16} - 3 \right)^{-1} \left( \log 2 + \frac{4K\alpha^2}{2 - \alpha^2} \right).$$

Finally, the next lemma suggests that if $\alpha$ is selected properly, then any $(\epsilon, \delta)$-reward free algorithm can be converted into an algorithm that provides the exact estimator with a probability of at least $1 - \delta$.

**Lemma D.3.** Suppose $\alpha \geq \frac{2\sqrt{2}\epsilon}{H-1}$, then any $(\epsilon, \delta)$-reward free algorithm could be converted to an algorithm which outputs an estimator $\widehat{\boldsymbol{\theta}}$, satisfying $\mathbb{E}_{\boldsymbol{\theta} \sim \mathrm{Unif}(\Theta)}[\mathbb{1}(\widehat{\boldsymbol{\theta}} = \boldsymbol{\theta})] \geq 1 - \delta$.

Equipped with these lemmas, we can provide the proof for Theorem 6.1.

*Proof of Theorem 6.1.* Set $\alpha = \frac{2\sqrt{2}\epsilon}{H-1}$ and $\epsilon \leq (H-1)/(2\sqrt{2})$, then by Lemma D.3, any $(\epsilon, \delta)$-reward free algorithm could be converted to an estimation algorithm with successful rate at least $1 - \delta$. Thus from Lemma D.2, the sample complexity $K$ of these reward free algorithms is bounded by

$$K \geq \frac{2 - \alpha^2}{4\alpha^2} \left( \frac{d-1}{16} - 3 \right)(1 - \delta) - \frac{2 - \alpha^2}{4\alpha^2} \log 2$$

$$\geq \frac{(H-1)^2}{128\epsilon^2} \left( \frac{d-1}{16} - 3 \right)(1 - \delta) - \frac{(H-1)^2 \log 2}{128\epsilon^2}.$$

Suppose $H \geq 2, d \geq 50, \delta \geq 1/2$ to simplify the result, we conclude that there exists an absolute positive constant $C$ such that $K \geq C(1 - \delta)H^2 d\epsilon^{-2}$, which leads to our final conclusion. $\square$

# E   Missing Proofs in Appendix D

We provide detailed proofs for lemmas in Appendix D. For simplicity, we denote by $d' = d - 1$ the dimension of the binary set $\mathcal{M}$.

## E.1   Proof of Lemma D.1

*Proof of Lemma D.1.* To begin with, we assume that $\mathbf{x} \sim \mathrm{Unif}(\mathcal{B}_{d'})$, i.e. $[\mathbf{x}]_i \sim \{-1, 1\}$. Thus given any $\mathbf{x}, \mathbf{x}' \sim \mathrm{Unif}(\mathcal{B}_{d'})$, we have

$$\mathbb{P}(\langle \mathbf{x}, \mathbf{x}' \rangle \geq d'\gamma) = \mathbb{P}_{z_i \sim \mathrm{Unif}\{-1,1\}} \left( \sum_{i=1}^{d'} z_i \geq d'\gamma \right)$$

$$= \mathbb{P}_{z_i \sim \mathrm{Unif}\{-1,1\}} \left( \frac{1}{d'} \sum_{i=1}^{d'} z_i \geq \gamma \right)$$

$$\leq \exp(-d'\gamma^2/2),$$

where the last inequality holds by utilizing the Azuma-Hoeffding's inequality with the fact that $z_i \sim \mathrm{Unif}\{-1, 1\}$ is a bounded random variable. Consider a set $\mathcal{M}$ with cardinality $|\mathcal{M}|$, then there is at most $|\mathcal{M}|^2$ pair of $(\mathbf{x}, \mathbf{x}')$. Thus taking a union bound over all vector pairs $(\mathbf{x}, \mathbf{x}')$, we have

$$\mathbb{P}(\exists \mathbf{x}, \mathbf{x}' \in \mathcal{M}, \mathbf{x} \neq \mathbf{x}', \langle \mathbf{x}, \mathbf{x}' \rangle \geq d'\gamma) \leq |\mathcal{M}|^2 \exp(-d'\gamma^2/2),$$

thus

$$\mathbb{P}(\forall \mathbf{x}, \mathbf{x}' \in \mathcal{M}, \mathbf{x} \neq \mathbf{x}', \langle \mathbf{x}, \mathbf{x}' \rangle \leq d'\gamma) \geq 1 - |\mathcal{M}|^2 \exp(-d'\gamma^2/2),$$

Once we have that $|\mathcal{M}|^2 \exp(-d'\gamma^2/2) < 1$, there exists a set $\mathcal{M}$ such that for any two different vector $\mathbf{x}, \mathbf{x}' \in \mathcal{M}$, $\langle \mathbf{x}, \mathbf{x}' \rangle \leq d\gamma$. $\square$

## E.2 Proof of Lemma D.2

We start our lower bound proof from Fano's inequality.

**Lemma E.1** (Fano's inequality, [6]). Consider probability measures $\mathbb{P}_{\boldsymbol{\theta}}, \boldsymbol{\theta} \in \boldsymbol{\Theta}$ on space $\Omega$ parameterized by $\boldsymbol{\theta} \in \boldsymbol{\Theta}$. Then for any estimator $\widehat{\boldsymbol{\theta}}$ on $\Omega$ and any comparison law $\mathbb{P}_0$ on $\Omega$

$$\frac{1}{|\boldsymbol{\Theta}|} \sum_{\boldsymbol{\theta} \in \boldsymbol{\Theta}} \mathbb{P}_{\boldsymbol{\theta}}[\widehat{\boldsymbol{\theta}} \neq \boldsymbol{\theta}] \geq 1 - \frac{\log 2 + \frac{1}{|\boldsymbol{\Theta}|} \sum_{\boldsymbol{\theta} \in \boldsymbol{\Theta}} \mathrm{KL}(\mathbb{P}_{\boldsymbol{\theta}}, \mathbb{P}_0)}{\log |\boldsymbol{\Theta}|}.$$

Then we will start our proof from the deterministic algorithms, which could be further extended to random algorithms using Yao's principle [23].

*Proof of Lemma D.2.* We denote $Y_k$ as such a trajectory at episode $k$ and $Y_{1:k}$ for the trajectories $Y_1, \cdots, Y_k$. We have for the KL divergence over joint distribution $Y_{1:k}$,

$$\begin{aligned}
&\mathrm{KL}(\mathbb{P}_{\boldsymbol{\theta}}(Y_{1:K}), \mathbb{P}_0(Y_{1:K})) \\
&= \sum_{Y_{1:K}} \mathbb{P}_{\boldsymbol{\theta}}(Y_{1:K}) \log(\mathbb{P}_{\boldsymbol{\theta}}(Y_{1:K})/\mathbb{P}_0(Y_{1:K})) \\
&= \sum_{Y_{1:K}} \mathbb{P}_{\boldsymbol{\theta}}(Y_{1:K-1})\mathbb{P}_{\boldsymbol{\theta}}(Y_K|Y_{1:K-1}) \log(\mathbb{P}_{\boldsymbol{\theta}}(Y_{1:K-1})/\mathbb{P}_0(Y_{1:K-1})) \\
&\quad + \sum_{Y_{1:K}} \mathbb{P}_{\boldsymbol{\theta}}(Y_{1:K-1})\mathbb{P}_{\boldsymbol{\theta}}(Y_K|Y_{1:K-1}) \log(\mathbb{P}_{\boldsymbol{\theta}}(Y_{1:K-1})/\mathbb{P}_0(Y_{1:K-1})) \\
&= \sum_{Y_{1:K-1}} \mathbb{P}_{\boldsymbol{\theta}}(Y_{1:K-1}) \log(\mathbb{P}_{\boldsymbol{\theta}}(Y_{1:K-1})/\mathbb{P}_0(Y_{1:K-1})) \sum_{Y_K} \mathbb{P}_{\boldsymbol{\theta}}(Y_K|Y_{1:K-1}) \\
&\quad + \sum_{Y_{1:K-1}} \mathbb{P}_{\boldsymbol{\theta}}(Y_{1:K-1}) \sum_{Y_k} \mathbb{P}_{\boldsymbol{\theta}}(Y_K|Y_{1:K-1}) \log(\mathbb{P}_{\boldsymbol{\theta}}(Y_{1:K-1})/\mathbb{P}_0(Y_{1:K-1})) \\
&= \mathrm{KL}(\mathbb{P}_{\boldsymbol{\theta}}(Y_{1:K-1}), \mathbb{P}_0(Y_{1:K-1})) + \mathbb{E}_{Y_{1:K-1}}[\mathrm{KL}(\mathbb{P}_{\boldsymbol{\theta}}(Y_K|Y_{1:K-1}), \mathbb{P}_0(Y_K|Y_{1:K-1})].
\end{aligned}$$

Thus by further expanding the above equations, we have

$$\mathrm{KL}(\mathbb{P}_{\boldsymbol{\theta}}(Y_{1:K}), \mathbb{P}_0(Y_{1:K})) = \sum_{k=1}^{K} \mathbb{E}_{Y_{1:k-1}}[\mathrm{KL}(\mathbb{P}_{\boldsymbol{\theta}}(Y_k|Y_{1:k-1}), \mathbb{P}_0(Y_k|Y_{1:k-1})],$$

where we denote $\mathbb{E}_{Y_{1:0}}[\mathrm{KL}(\mathbb{P}_{\boldsymbol{\theta}}(Y_1|1:0), \mathbb{P}_0(Y_1|1:0)] := \mathrm{KL}(\mathbb{P}_{\boldsymbol{\theta}}(Y_1), \mathbb{P}_0(Y_0))$ for consistency.

Since for any deterministic algorithm, in any episode, the trajectory $s_1, a_1, \cdots, s_H, a_H$ is determined after the algorithm goes into $S_{2,1}$ or $S_{2,2}$, furthermore, for these deterministic algorithms, the first action $a$ at $k$-th trajectory is fixed given previous knowledge $Y_{1:k}$. Therefore, the distribution of the whole trajectory could be replaced by the distribution of $S_{2,1}$ and $S_{2,2}$. We have there are at most two possible value for $Y_k$, we denote the trajectory $S_1, a_1, S_{2,1}, \cdots, S_{2,1}$ by $Y_k = 0$ and the other trajectory $S_1, a_1, S_{2,2}, \cdots, S_{2,2}$ by $Y_k = 1$. We define the comparison distribution $\mathbb{P}_0$ as

$$\mathbb{P}_0(Y_k = 0|Y_{1:k-1}) = \frac{1}{|\boldsymbol{\Theta}|} \sum_{\boldsymbol{\theta}_i \in \boldsymbol{\Theta}} \mathbb{P}_{\boldsymbol{\theta}_i}(Y_k = 0|Y_{1:k-1}) := \frac{1}{2} + \frac{\alpha}{\sqrt{2d'}} \langle \mathbf{a}, \bar{\boldsymbol{\theta}} \rangle$$

$$\mathbb{P}_0(Y_k = 1|Y_{1:k-1}) = \frac{1}{|\boldsymbol{\Theta}|} \sum_{\boldsymbol{\theta}_i \in \boldsymbol{\Theta}} \mathbb{P}_{\boldsymbol{\theta}_i}(Y_k = 1|Y_{1:k-1}) := \frac{1}{2} - \frac{\alpha}{\sqrt{2d'}} \langle \mathbf{a}, \bar{\boldsymbol{\theta}} \rangle,$$

where we denote $\bar{\boldsymbol{\theta}}$ is the mean value of $\widetilde{\boldsymbol{\theta}}_i \in \mathcal{M}$. (Recall that $\boldsymbol{\Theta} = \{\boldsymbol{\theta}_i | \boldsymbol{\theta}_i = (\sqrt{2}, \alpha \widetilde{\boldsymbol{\theta}}_i^\top / \sqrt{d'})^\top \}$.) For simplicity, we use $\mathbb{P}_{\boldsymbol{\theta}_i}$ and $\mathbb{P}_0$ to denote the distributions for the whole trajectory defined above.

Then we could bound the KL divergence between $\mathbb{P}_0$ and $\mathbb{P}_{\boldsymbol{\theta}_i}$ as

$$\mathrm{KL}(\mathbb{P}_{\boldsymbol{\theta}_i}, \mathbb{P}_0)$$

$$= \left( \frac{1}{2} + \frac{\alpha}{\sqrt{2}d'} \langle \mathbf{a}, \widetilde{\boldsymbol{\theta}}_i \rangle \right) \log \left( \frac{\sqrt{2}d' + \alpha \langle \mathbf{a}, \widetilde{\boldsymbol{\theta}}_i \rangle}{\sqrt{2}d' + \alpha \langle \mathbf{a}, \bar{\boldsymbol{\theta}} \rangle} \right)$$

$$+ \left( \frac{1}{2} - \frac{\alpha}{\sqrt{2}d'} \langle \mathbf{a}, \widetilde{\boldsymbol{\theta}}_i \rangle \right) \log \left( \frac{\sqrt{2}d' - \alpha \langle \mathbf{a}, \widetilde{\boldsymbol{\theta}}_i \rangle}{\sqrt{2}d' - \alpha \langle \mathbf{a}, \bar{\boldsymbol{\theta}} \rangle} \right)$$

$$\leq \left( \frac{1}{2} + \frac{\alpha}{\sqrt{2}d'} \langle \mathbf{a}, \widetilde{\boldsymbol{\theta}}_i \rangle \right) \frac{\alpha \langle \mathbf{a}, \widetilde{\boldsymbol{\theta}}_i - \bar{\boldsymbol{\theta}} \rangle}{\sqrt{2}d' + \alpha \langle \mathbf{a}, \bar{\boldsymbol{\theta}} \rangle} - \left( \frac{1}{2} - \frac{\alpha}{\sqrt{2}d'} \langle \mathbf{a}, \widetilde{\boldsymbol{\theta}}_i \rangle \right) \frac{\alpha \langle \mathbf{a}, \widetilde{\boldsymbol{\theta}}_i - \bar{\boldsymbol{\theta}} \rangle}{\sqrt{2}d' - \alpha \langle \mathbf{a}, \bar{\boldsymbol{\theta}} \rangle}.$$

Taking summation over $\boldsymbol{\theta}_i \in \boldsymbol{\Theta}$, recall that $\bar{\boldsymbol{\theta}}$ is the mean value of $\widetilde{\boldsymbol{\theta}}_i \in \mathcal{M}$, we have

$$\sum_{\boldsymbol{\theta}_i \in \boldsymbol{\Theta}} \mathrm{KL}(\mathbb{P}_{\boldsymbol{\theta}_i}, \mathbb{P}_0) = \frac{2\alpha^2}{2d'^2 - \alpha^2 \langle \mathbf{a}, \bar{\boldsymbol{\theta}} \rangle^2} \sum_{\boldsymbol{\theta}_i \in \boldsymbol{\Theta}} \langle \mathbf{a}, \widetilde{\boldsymbol{\theta}}_i \rangle \langle \mathbf{a}, \widetilde{\boldsymbol{\theta}}_i - \bar{\boldsymbol{\theta}} \rangle.$$

Given the fact that $\langle \mathbf{x}, \mathbf{x}' \rangle \leq d'$ for any $\mathbf{x}, \mathbf{x}' \leq \mathcal{A}$, one can easily get that

$$\frac{1}{|\boldsymbol{\Theta}|} \sum_{\boldsymbol{\theta}_i \in \boldsymbol{\Theta}} \mathrm{KL}(\mathbb{P}_{\boldsymbol{\theta}_i}, \mathbb{P}_0) \leq \frac{4\alpha^2}{2 - \alpha^2}.$$

Plugging the above inequality into the decomposition of KL divergence, from Fano's inequality Lemma E.1, we have

$$\delta \geq \frac{1}{|\boldsymbol{\Theta}|} \sum_{\boldsymbol{\theta} \in \boldsymbol{\Theta}} \mathbb{P}_{\boldsymbol{\theta}}[\widehat{\boldsymbol{\theta}} \neq \boldsymbol{\theta}] \geq 1 - \left( \frac{d'}{16} - 3 \right)^{-1} \left( \log 2 + \frac{4K\alpha^2}{2 - \alpha^2} \right).$$

Replacing $d'$ by $d - 1$, we can get the same result as the statement of the lemma. $\qquad \square$

### E.3 Proof of Lemma D.3

We show the proof for Lemma D.3 by establishing different reward functions for this MDP structure.

*Proof of Lemma D.3.* For any $\boldsymbol{\theta} \in \boldsymbol{\Theta}$, we build the reward sequence as $r(S_1) = r(S_{2,1}) = 0, r(S_{2,2}) = 1$, then any $(\epsilon, \delta)$-correct algorithm guarantees that

$$\mathbb{P}(V^*(S_1, r; \boldsymbol{\theta}) - V^\pi(S_1, r; \boldsymbol{\theta}) \leq \epsilon) \geq 1 - \delta, \forall \boldsymbol{\theta} \in \boldsymbol{\Theta}.$$

Our proof is to show that, as long as $\alpha \geq \frac{2\sqrt{2}\epsilon}{H-1}$, we can build up the estimation of $\boldsymbol{\theta}$ using $\boldsymbol{\theta}_i = \left( \sqrt{2}, \alpha \mathbf{a}_i^\top / \sqrt{d} \right)^\top$ where $\mathbf{a}_i$ is determined by $a_i = \pi(S_1)$. It is guaranteed that $\mathbb{P}_{\boldsymbol{\theta}}[\boldsymbol{\theta}_i = \boldsymbol{\theta}] \geq 1 - \delta$ and furthermore, $\mathbb{E}_{\boldsymbol{\theta} \sim \mathrm{Unif}(\boldsymbol{\Theta})}(\mathbb{1}[\boldsymbol{\theta} = \widehat{\boldsymbol{\theta}}]) \geq 1 - \delta$.

Suppose for the MDP parameter $\boldsymbol{\theta}_i$, it is easy to find that the optimal policy for the first step is $\pi^*(S_1) = a_i$. Suppose that for any policy $\pi(S_1) = a_j$ where $j \neq i$, then from the MDP structure, the gap between policy and the optimal policy is

$$V^*(S_1, r; \boldsymbol{\theta}_i) - V^\pi(S_1, r; \boldsymbol{\theta}_i) = \frac{(H-1)\alpha}{\sqrt{2}d} (\langle \mathbf{a}_i, \mathbf{a}_i \rangle - \langle \mathbf{a}_i, \mathbf{a}_j \rangle)$$

$$\geq \frac{(H-1)\alpha}{\sqrt{2}d'} (d' - d'/2)$$

$$= \frac{(H-1)\alpha}{2\sqrt{2}},$$

as long as we have $\alpha \geq \frac{2\sqrt{2}\epsilon}{H-1}$, we can get the policy gap $V^*(S_1, r; \boldsymbol{\theta}_i) - V^\pi(S_1, r; \boldsymbol{\theta}_i) \geq \epsilon$.

Therefore, it is easy to show that the estimation using the policy $\pi(S_1)$ is guaranteed with successful rate at least $1 - \delta$ for any MDP parameter $\boldsymbol{\theta}_i$, thus we can further conclude that $\mathbb{E}_{\boldsymbol{\theta} \sim \mathrm{Unif}(\boldsymbol{\Theta})}(\mathbb{1}[\boldsymbol{\theta} = \widehat{\boldsymbol{\theta}}]) \geq 1 - \delta$. $\qquad \square$