# OpenReview forum: "Reward-Free Model-Based Reinforcement Learning with Linear Function Approximation"
_NeurIPS.cc/2021/Conference — NeurIPS 2021 Poster_

### Official Review · Reviewer_6DWQ · 2021-07-13

**Rating:** 6
**Confidence:** 4

**Summary:**

This paper considers the Reward Free Reinforcement Learning for the linear mixture MDP assumption. Their (model-based) UCRL-RFE achieves $\tilde{O}(H^5d^2\epsilon^{-2})$ complexity and can be further improved to $\tilde{O}(H^4d(H+d)\epsilon^{-2})$ with a Bernstein-type design. In addition, it also provides a lower bound for reward-free RL with $\Omega(H^2d\epsilon^{-2})$ in the linear mixture MDP setting.

**Ethical Concerns:**

No.

**Limitations And Societal Impact:**

Yes.

**Main Review:**

In general, this paper provides some nice contributions to the reward-free RL with the best finite sample guarantee of order $\tilde{O}(H^4d(H+d)\epsilon^{-2})$. However, there are a few reasons that make the current version slightly below the acceptance bar in my opinion.

 The lower bound obtained in Theorem 6.1 is not tight, which makes the discussion of matching parameter dependence vague. Concretely, it is known that reward-free learning is harder than the standard reward-specific task. For the linear mixture MDP, the reward-specific task is known to have the regret lower bound $\Omega(dH\sqrt{T})$ (ZHOU, D., GU, Q. and SZEPESVARI, C. COLT 21). This will be translated into $\Omega(H^3d^2\epsilon^{-2})$ in the PAC sense. Therefore, the tight lower bound should be of order at least $d^2$.

Hence, the claim "Our upper bound matches the lower bound in terms of the dependence on ε and the dependence on d if $H\geq d$" is vague since the dependence on $d$ should at least be $d^2$ instead of $d$. Also, the way to do the claim is lacking rigor since $\tilde{O}(H^4d(H+d)\epsilon^{-2})\geq \tilde{O}(H^4d^2\epsilon^{-2})$, the dependence is always quadratic. Such a weak lower bound diminishes the contribution of UCRL-RFE since it is not clear how far your upper bound is from the "true" statistical limits of the current problem. Indeed, even in the tabular offline RL, the PAC bound for outputting a $\epsilon$-optimal policy for the reward-free task has a lower bound $\Omega(H^2S/d_m\epsilon^{2})$ (which translates to quadratic dependence in $S$ https://arxiv.org/pdf/2105.06029.pdf). Understanding the correct (statistical) rate for the setting is important.

Lastly, your remark 3.2 states "linear mixture MDPs do not need the reward function $r$ to be linear, which makes our algorithms more general." Indeed, it is commonly known that for linear MDP and linear mixture MDP, one does not contain the other. Therefore, I am not sure what the word "general" means here.

Overall, the paper provides a nice perspective and by further modification, the paper's quality could increase. Therefore, I choose the weak reject.







**Time Spent Reviewing:**

10 Hours

---

> ### Author Response · Authors · 2021-08-10
> **Response to Reviewer 6DWQ**
>
> Thank you for your constructive feedback. Here’s our response to your comments.
>
> **Q1**: The lower bound part is loose, the tight lower bound should be of order at least $d^2$
>
> **A1**: We are a bit skeptical about the claim that ‘the lower bound should be at least $d^2$. Consider PAC bounds for the best-arm identification problem in linear bandits. For a $d$-dimensional, $K$-armed linear bandits problem, the best existing lower bound of sample complexity to find an $\epsilon$-suboptimal arm is $\tilde \Omega(\epsilon^{-2}\cdot d)$ according to [1,2]. However, the best known lower bound of the regret for $d$-dimensional bandits is $\Omega(d\sqrt{T})$, which somehow contradicts your claim that we can directly ‘transfer’ a regret bound to a sample complexity bound. Currently, we tend to believe that the upper bound is improvable.
>
> [1] Soare, Marta, Alessandro Lazaric, and Rémi Munos. "Best-arm identification in linear bandits." Advances in Neural Information Processing Systems 27 (2014): 828-836.
>
> [2] Chen, Jiecao, et al. "Adaptive multiple-arm identification." International Conference on Machine Learning. PMLR, 2017.
>
> ---
>
> **Q2**: ‘’...in the tabular offline RL, the PAC bound for outputting a ϵ-optimal policy for the reward-free task has a lower bound $\Omega(H^2S/d_m\epsilon^2)$ ... Understanding the correct (statistical) rate for the setting is important.”
>
> **A2**. Thank you for pointing out the relevant paper. We will comment on it in the revision.
>
> ---
>
> **Q3**: Linear Mixture MDP and Linear MDP does not contain each other
>
> **A3**: We agree. We will rephrase the sentence to be “linear mixture MDPs do not need the reward function r to be linear, which makes our algorithms can deal with general reward functions.”

---

> > ### Author Response · Authors · 2021-08-30
> > **Followup**
> >
> > Thank you again for your comments! We believe our response has addressed all your comments and questions. Please let us know if you have any other questions, and we are happy to discuss more. If you’re satisfied with our response, we sincerely hope you could reconsider the rating.

---

### Official Review · Reviewer_USFu · 2021-07-15

**Rating:** 6
**Confidence:** 4

**Summary:**

Reward free learning is a recently proposed reinforcement learning (RL) problem which has seen significant attention in the RL community. At the same time, the RL community has begun to devote much attention to obtaining theoretical results for MDPs with linear structure. This works seeks to bridge these two settings, and studies reward free learning in linear MDPs (in particular, the linear mixture MDP setting). They propose and analyze two algorithms based on the UCRL algorithm, and show that they attain sample complexities close to minimax optimal.

**Limitations And Societal Impact:**

The authors do not discuss potential societal impact. However, the work is primarily theoretical so it is difficult to determine what negative societal impacts it may have, if any. Limitations are discussed.

**Main Review:**

Reward free exploration (RFE) is a relatively new problem in the RL community where the goal is for the agent to explore an MDP so that it can derive a near optimal for any reward function. While minimax optimal algorithms now exist for this problem in the tabular setting, less attention has been given to reward free learning in the setting of linear MDPs. This work expands on this literature and studies reward free learning in linear MDPs (in particular, in the linear mixture MDP setting of [1]).

The authors propose two algorithms, both based on the UCRL algorithm. The first, which relies on Hoeffding bonuses, achieves a sample complexity of O(H^5 d^2/epsilon^2). The second, which relies on Bernstein bonuses, achieves a sample complexity of O(H^4 d (d+H)/epsilon^2). In their original form, neither algorithm is computationally efficient, but the authors propose a computationally efficient version which achieves a slightly worse complexity of O(H^5 d^3/epsilon^2), though it requires that the number of states be finite. In addition, the authors prove a lower bound of \Omega(H^2 d/epsilon^2) for reward free learning in the linear mixture MDP setting.

Pros:
- Reward free learning and learning in linear MDPs have both been topics of significant recent interest in the RL community but only a handful of works have studied reward free learning in linear MDPs. This work, then, fills a gap in the existing literature and addresses a relevant problem. In particular, it is the first work to study RFE in the linear mixture MDP setting of [1].
- To my knowledge, this is the first algorithm for RFE in MDPs with some form of linear structure that achieves the minimax bound in terms of the dimensionality. Previous works have only achieved complexity scaling as d^3. As such, this is the state-of-the-art bound for RFE in MDPs with linear structure (though it is important to note that it only achieves this when H >= d).
- The lower bound is, I believe, the first lower bound for reward-free learning with linear function approximation.

Cons:
- The algorithms proposed are not computationally efficient. While they do present a computationally efficient relaxation, this only applies in the setting where the number of states is finite (i.e. the setting where linear function approximation is not typically necessary), and does not achieve the optimal rates (the computationally efficient versions both have a rate of O(H^5 d^3/\epsilon^2)).
- Their best algorithm (UCRL-RFE+) does not hit the optimal dependence on H and only hits the optimal dependence on d in the case when H >= d. While this is the state of the art sample complexity for linear function approximation, it is therefore still not optimal.
- [2] and [3] should be cited and mentioned.

This work is certainly a contribution in that it solves an unsolved but timely problem. However, while it improves on the state of the art, it does not hit the optimal rate and is not computationally efficient. As such, I believe it is a borderline accept—it makes a contribution but is not ground-breaking.


[1] Ayoub, Alex, et al. "Model-based reinforcement learning with value-targeted regression." International Conference on Machine Learning. PMLR, 2020.
[2] Agarwal, Alekh, et al. "Flambe: Structural complexity and representation learning of low rank mdps." arXiv preprint arXiv:2006.10814 (2020).
[3] Qiu, Shuang, et al. "On Reward-Free RL with Kernel and Neural Function Approximations: Single-Agent MDP and Markov Game." International Conference on Machine Learning. PMLR, 2021.


**Time Spent Reviewing:**

3

---

> ### Author Response · Authors · 2021-08-10
> **Response to Reviewer USFu**
>
> Thank you for your positive feedback. Here’s our response to your concerns regarding the paper.
>
> **Q1**: The algorithms proposed are not computationally efficient
>
> **A1**: The proposed algorithm is computationally efficient when the state space is finite. When the state space is infinite, we need to use epsilon-net to cover the continuous state space, and reduce the state space from infinite to finite. Note that this is not the drawback of our algorithm, but the drawback of linear mixture MDPs (the feature mapping $\phi_f(s, a)$ is defined as the integration of the original feature mapping $\phi(s, a)$ with respect to a certain function $f$).
>
> ---
>
> **Q2**: “While this is the state of the art sample complexity for linear function approximation, it is therefore still not optimal.”
>
> **A2**: As you acknowledged, even though our result is still not optimal, it is state-of-the-art. We hope our work can inspire follow-up work to close this gap.
>
> ---
>
> **Q3**: [2] and [3] should be mentioned and cited
>
> **A3**: Thanks for pointing out those papers, we will comment on them in the related work section.

---

> ### Comment · Reviewer_USFu · 2021-08-18
> **Updated review**
>
> After reading the authors' comments and the other reviews, my opinion of this paper has not changed and I would like to maintain my score of 6.

---

### Official Review · Reviewer_nnra · 2021-07-16

**Rating:** 6
**Confidence:** 4

**Summary:**

The authors study the problem of reward free exploration in the Linear Mixture MDP model. They propose the UCRL-RFE algorithm and show the number of samples required during the exploration phase is at most of the order of O(H^5d^2epsilon^{-2}). They also develop a "fast" version of their algorithm/analysis that shows a rate of O(H^4 d(H+d)epsilon^{-2}). They complement these results with a lower bound showing they are unimprovable in their dependence on d and epsilon in case H \geq d.


**Ethical Concerns:**

No ethical concerns.

**Limitations And Societal Impact:**

Adequate

**Main Review:**

First comment, the paper's name is awfully similar to the related work
The paper's name is extremely similar to: "On Reward-Free Reinforcement Learning with Linear Function Approximation". It may be a good strategy to change the name of the paper (if possible) to better reflect the contents of this work and draw distinctions with previous work.

Originality

This is yet another paper treating the problem of reward free exploration in MDPs. The linear approximation model that the authors work with (the MDP linear mixture model) is different from that discussed in previous works in the topic such as the aforementioned paper. This work different from the reward free approaches to Linear MDPs in that it is a model based algorithm more related to "Reward-Free Exploration for Reinforcement Learning".  The results are technically correct but it is unclear if they have any component of unexpected in them.

Quality and clarity

The paper is well written and structured.

Significance

The results probably meet the bar for publication in the conference. The results are not very surprising and the problem studied in this work is an obvious extension of existing results, but it may be worthwhile showcasing them.






**Time Spent Reviewing:**

2 hrs

---

> ### Author Response · Authors · 2021-08-10
> **Response to Reviewer nnra**
>
> Thank you for your positive comments.
>
> **Q1**: Change the paper title
>
> **A1**: Thank you for your suggestion. The keyword in our current title is “Model-based”, in contrast to the model-free algorithm proposed in the paper “On Reward-Free Reinforcement Learning with Linear Function Approximation”. We will consider changing our title to make it reflect the key technique of our algorithm, such as the pseudo value function.

---

> > ### Comment · Reviewer_nnra · 2021-08-30
> > **thanks for your reply**
> >
> > Thanks for your reply. I will keep my score.

---

### Official Review · Reviewer_Zh4L · 2021-07-17

**Rating:** 6
**Confidence:** 3

**Summary:**

This paper studies the problem of reward-free exploration in linear mixture MDPs. In the reward-free setting, the agent interacts in two stages where it must explore sufficiently without a reward function and then an arbitrary function is revealed that it must now solve without further interaction. Two reward-free algorithms based on UCRL are proposed. Both algorithms aim to reduce uncertainty. The second is an improvement that leverages variance estimates to improve the regret bound in terms of H. A lower bound is proved, but it is not matching.

**Limitations And Societal Impact:**

Limitations are discussed in the main review.

**Main Review:**

The paper offers some interesting results in the reward-free setting for linear mixture MDPs, which answer questions about sample complexity for this type of model. It is interesting that one is able to achieve d^2/eps^{-2} in this model (better than other models, but matching what is achievable in the non-reward free setting). The paper is very well-written and easy to understand. The related work also does a good job of distinguishing the work from past work in both reward-free learning and general reinforcement learning with linear function approximation.

I do not have any technical complaints about the paper. It matches assumptions that have been made in the past without imposing more. The analysis seems technically correct from what I can tell and it is grounded in some fairly tried-and-true techniques.

My primary critique is that the paper does not seem to introduce any technically novel analysis, approach, model, or problem setting. It seems like a relatively simple combination of existing ideas, which is not inherently bad, but it also yielded an unsurprising conclusion. By now, there are many existing papers that have already shown that essentially applying standard online algorithms with good exploration is a good strategy for reward-free learning. So, while I think this is a good paper that answers some questions that definitely should be answered, the significance and novelty of the contribution seems to be lacking.


**Time Spent Reviewing:**

4

---

> ### Author Response · Authors · 2021-08-10
> **Response to Reviewer Zh4L**
>
> We sincerely thank the reviewer for the comments.
>
> **Q1**: “the paper does not seem to introduce any technically novel analysis,”
>
> **A1**: We respectfully disagree. In fact, we designed the pseudo value function $u_h^k$ and used it to guarantee the provable statistical efficiency of our algorithm. This kind of pseudo value function has never appeared in previous work. Moreover, in Algorithm 3, to use the Bernstein-style bonus, we not only construct estimators $\hat \theta, \check \theta$ and $\tilde \theta$ by solving the regression problem over the value function $V_{h+1}^k$, but also construct estimator $\theta$ over the pseudo value function $u_h^k$. This additional estimator is quite different from the ones in existing literature and it indeed brings in additional challenges in the analysis. Finally, regarding the lower bound part, our construction of the ‘hard instances’ is new, and the proof of the lower bound is also novel and has never appeared in prior work.

---

> > ### Author Response · Authors · 2021-08-30
> > **Followup**
> >
> > Thank you again for your comments! We believe our response has addressed your concerns. Please let us know if you have any other questions, and we are happy to discuss more.

---

> > ### Author Response · Authors · 2021-09-02
> > **We would appreciate an update from you**
> >
> > Thank you again for your time and efforts in providing the review.
> >
> > We would like to make sure that our response has addressed your concerns regarding the technical novelty of our paper. We would also appreciate it if you can let us know if you have any other questions/concerns, and we are happy to further address them. We sincerely hope you could reconsider the rating as the other reviewers are all satisfied with our response.

---

> > ### Comment · Reviewer_Zh4L · 2021-09-13
> > **Response**
> >
> > Thank you for your reply. After the discussion and re-reading, I have updated the score.

---

### Decision · Program_Chairs · 2021-09-28

**Decision:**

Accept (Poster)

**Comment:**

This paper proposes two algorithms (corresponding to Bernstein and Hoeffding bonuses) for reward-free exploration in linear mixture MDPs and analyzes their sample complexity. The authors also provide a lower bound for this setting. The proposed algorithms involve a computationally difficult subproblem, which is replaced by an LP relaxation in the implementation.
Overall, the results and techniques are not very surprising, leaving most reviewers ambivalent. However, given that the paper is executed well and fills a gap in the literature, I recommend acceptance.


**Consistency Experiment:**

NeurIPS has a long history of experimentation. In 2014, NeurIPS ran an experiment in which 10% of submissions were reviewed by two independent committees to quantify the randomness in the review process. This year, we repeated a variant of this experiment to see how the quality of the review process has changed over time.  This paper was part of the experiment and was therefore assigned to two committees (consisting of reviewers, an Area Chair, and a Senior Area Chair) that reached independent decisions.  If both committees made the same recommendation, this recommendation was followed. If a single committee recommended acceptance, the paper was accepted (with the exception of a few cases in which the other committee identified what we considered a fatal flaw, e.g., an error in a key result).

This copy’s committee reached the following decision: **Accept (Poster)**

The other committee assigned to the paper recommended **Reject**.  You can find the other set of reviews, along with any follow up discussion with the authors here:
https://openreview.net/forum?id=YsGQImFVkoN